

**Comparative Analysis of MODIS, MISR and AERONET Climatology**
**over the Middle East and North Africa**
**Ashraf Farahat**
Department of Physics, King Fahd University of Petroleum and Minerals, Dhahran 31261,
Saudi Arabia;
E-Mails: farahata@kfupm.edu.sa
*Author to whom correspondence should be addressed; E-Mail: farahata@kfupm.edu.sa.
Tel: (321) 541-7088
**Abstract:**
Comparative analysis of MISR MODIS, and AERONET AOD products is performed over
seven AERONET stations located in the Middle East and North Africa for the period of
2000 – 2015. Sites are categorized into dust, biomass burning and mixed. MISR and
MODIS AOD agree during high dust seasons but MODIS tends to underestimate AOD
during low dust seasons. Over dust dominated sites, MODIS/Terra AOD indicate a
negative trend over the time series, while MODIS/Aqua, MISR, and AERONET depict a
positive trend. A deviation between MODIS/Aqua and MODIS/Terra was observed
regardless of the geographic location and data sampling. The performance of MODIS is
similar over the entire region with ~68% of AOD within the $\Delta\tau = \pm0.05 \pm 0.15\tau_{AERO}$
confidence range. MISR AOD retrievals fall within 72% of the same confidence range for
all sites examined here. Both MISR and MODIS capture aerosol climatology; however few
cases were observed where one of the two sensors better captures the climatology over a
certain location or AOD range than the other sensor.
**Keywords:** AOD; Remote Sensing; North Africa; Middle East; Validation



## 1. Introduction

The Middle East and North Africa host the largest dust source in the world, the Sahara
Desert in North Africa that may be responsible for up to 18 percent of global dust emission
(Todd et al., 2007, Bou Karam et al. 2010, Schepanski et al. 2016). The vast 650,000 km$^2$
Rub' al Khali (Empty Quarter) sand desert is a major source of frequent dust outbreaks and
severe dust storms that has major effect on human activity in the Arabian Peninsula (Böer,
1997, Elagib and Addin 1997, Farahat et al., 2015).
Air quality over the Arabian Peninsula has received significant attention during the past 15
years due to *unprecedented overall economic growth, and a booming oil and gas industry,*
*however, air pollution studies are still far from complete.* Frequently blowing dust storms
play a significant role in pollutant transport over the Arabian Peninsula; and major
environmental pollution events such as burning of Kuwait oil fields during the 1991, Gulf
War resulted in a large environmental impact on the Arabian Gulf Area (Sadiq and
McCain, 1993, and Farahat 2016).
Aerosol optical depth, AOD, (also called aerosol optical thickness, AOT) as a parameter
indicates the extinction of a beam of radiation as it passes through a layer of atmosphere
that contains aerosols. Both satellites and ground-based instruments can be used to measure
AOD in the atmosphere, but within the same temporal coordinates and geographic location
different instruments could generate different retrievals (Kahn et al., 2007, Kokhanovsky et
al., 2007, Liu et al., 2008 and Mishchenko et al., 2009).
Since the turn of the 21$^{st}$ century, an upward trend of remotely sensed and ground-based
AOD and air pollutants was observed over the Middle East and North Africa (El-Askary
2009, Ansmann et al. 2011, Yu et al. 2013, Chin et al. 2014, Yu et al. 2015, Farahat et al.
2016, Solomos et al. 2017). This positive trend is attributed to the increase in the Middle
Eastern dust activity (Hsu et al., 2012) due to changes in wind speed and soil moisture
(Ginoux et al. 2001 and Kim et al. 2013). Yu et al., (2015) concluded that the persistent La



Niña conditions (Hoell et al., 2013) have caused increment in Saudi Arabian dust activity
during 2008 – 2012.  Energy subsidies also encourages energy overconsumption in the
Middle East and North Africa with little incentive to adopt cleaner technology. Lack of
applying strict environmental regulations have permitted exacerbated urban air pollution.
During the last two decades, a large number of satellites, ground stations and computational
models contributed to build global and regional maps for the temporal and spatial aerosol
distributions. While, ground-based stations and field measurements can identify aerosols
properties over specific geographic locations, the spare and non-continues data from
ground-based sensors scattered over the Middle East and North Africa is not sufficient to
provide information on spatial and temporal trends of particulate pollution. On the other
hand, satellites imagery could provide a significant source of data mapping over larger
areas.
For its wide spatial and temporal data availability space-born sensors are important sources
to understand aerosols characteristics and transport, however low sensitivity to particle type
under some physical conditions, high surface reflectivity, persistent cloud, and generally
low aerosol optical depth could limit satellite data application in characterizing properties
of airborne particles, especially in the Middle East.
In order to evaluate the efficiency of space-borne sensors in representing ground observations
recorded by AERONET stations we have performed detailed statistical inter-comparison
analysis between satellite AOD products and AERONET for seven stations in the Middle East
and North Africa representative for dust, biomass burning, and mixed aerosol conditions
(Dubovik et al., (2000, 2002, 2006), Holben et al. (2001), Derimian et al., (2006), Basart et
al. (2009), Eck el. (2010), Marey et al., 2010, Abdi et al., (2012)). Previously we analysed
these seven AERONET stations to understand particles categorization and absorption
properties (Farahat et al. 2016), and the current study extends the analysis to the satellite
datasets.



In the first part of this article, we validated MISR and MODIS retrievals against collocated
AERONET observations. We also assessed the consistency in aerosol trends between
space-borne sensors and ground-based data.
In the second part, we evaluated representativeness of satellite-derived aerosol climatology
over the study region from the long-term AERONET data for MISR and MODIS AOD
products. It is especially relevant for the MISR instrument, as its sampling is limited by
once per week observations of the same region from the two overlapping paths.  MODIS
provides nearly daily observations to the same geographic location; however, the quality of
the product diminishes over the bright targets potentially affecting MODIS-derived aerosol
climatology.
The collocated MISR, MODIS and AERONET data were obtained at the MAPSS website
(http://giovanni.gsfc.nasa.gov/mapss.html).

**2. Materials and Methods**
**2.1 MISR**
The Multi-angle Imaging SpectroRadimeter (MISR) instrument to measures tropospheric
aerosol characteristics through the acquisition of global multi-angle imagery on the
daylight side of Earth. MISR applies nine Charge Coupled Devices (CCDs), each with 4
independent line arrays positioned at nine view angles spread out at nadir, $26.1^{o}$, $45.6^{o}$,
$60.0^{o}$, and $70.5^{o}$. In each of the nine MISR cameras, images are obtained from reflected and
scattered sunlight in 4 bands blue, green, red, and near-infrared with a centre wavelength
value of 446, 558, 672, and 867 nm respectively. The combination of viewing cameras and
spectral wavelengths enables MISR to retrieve aerosols AOD over high reflection surfaces
like deserts.
In this study, we use Level 2 (ver. 0022) AOD at 558 nm (green band) measured by MISR
instrument with a 17.6 km resolution aboard the Terra satellite. MISR Level 2 aerosol



retrievals use only data that pass angle-to-angle smoothness and spatial correlation tests
(Martonchik et al. 2002), as well as stereoscopically derived cloud masks and adaptive
cloud-screening brightness thresholds (Zhao and Di Girolamo, 2004).
**2.2 MODIS**
The Moderate Resolution Imaging Spectroradiometer (MODIS) is a payload instrument on
board the Terra and Aqua satellites. Terra's and Aqua orbit around the Earth from North to
South and South to North across the equator during the morning and afternoon respectively
(Kaufman et al., 1997). Terra MODIS and Aqua MODIS provides nearly daily coverage of
the Earth's surface and atmosphere in 36 wavelength bands, ranging from 0.412 to 41.2
μm, with spatial resolutions of 250 m (bands 1-2), 500 m (bands 3-7), 1000 m (bands 8-
36). Located near-polar orbit (705 km), MODIS has swath dimensions of 2330 km × 10 km
and a scan rate of 20.3 rpm.  With its high radiometric sensitivity and swath resolution
MODIS retrievals provides information about aerosols optical and physical characteristics.
MODIS uses 14 spectral band radiance values to evaluate atmospheric contamination and
determine whether scenes are affected by cloud shadow (Ackerman et al., 1998).
The MODIS dark-target algorithm is designed aerosol retrieval from MODIS observations,
over dark land surfaces (low values of surface reflectance) (e.g., dark soil and vegetated
regions) in parts of the visible (VIS, 0.47 and 0.65 μm) and shortwave infrared (SWIR, 2.1
μm) spectrum (Kaufman et al., 1997). Level 2 (C006) of the algorithm are used to retrieve
MODIS aerosols' time series data. Levy *et al.* (2010) reported that the dark-target
algorithm AOD at 550 nm measurement for (C005) includes uncertainty of $\pm (0.05\tau + 0.03)$
and $\pm (0.15\tau + 0.05)$ over ocean and land respectively. This uncertainty is caused by
uncertainties in computing cloud masking, surface reflectance, aerosol model type (e.g.,
single scattering albedo), pixels selections and instrument calibration.





### 2.3 AERONET


The Aerosol Robotic Network (AERONET) (Holben et al., 1998 and Holben et al., 2001) is
a ground-based remote sensing aerosols network that provides a long-term data related to
aerosol optical, microphysical and radiative properties. With over 700 global stations, the
AERONET data is widely used in validating satellite retrievals (Chu et al., 1998 and
Higurashi et al., 2000).
The sun photometers used by AERONET measure spectral direct-beam solar radiation, as
well as directional diffuse radiation in the solar almucantar. The former are used to
determine columnar spectral AOD and water vapour, provided at a temporal resolution of
approximately 10–15 min (Sayer et al. 2014). AERONET direct-sun AOD has a typical
uncertainty of 0.01–0.02 (Holben et al., 1998) and is provided at multiple wavelengths at
340, 380, 440, 500, 675, 950, and 1020 nm.
Seven AERONET sites were selected for satellite validation in this study (Table 1.). The
sites were selected based on their geographic locations to represent aerosols characteristics
over North Africa and the Middle East (Farahat et al., 2016). A record of long-term data
collection was another factor in the selection process.

### Data Matching Approach


Multi-sensors data matching requires using only compatible data to eliminate uncertainties
associated with cloud shadow and spatial and temporal retrievals produced by different
instruments (Liu and Mishchenko (2008) and Mishchenko et al., 2009).
The comparison of MISR and MODIS products against AERONET is performed to
evaluate satellites' retrieval over individual North Africa and Middle East sites (see Table
1). There is only a small number of AERONET measurements that are perfectly collocated
with MODIS and MISR. One way to work with this lack of compatibility problem is to
compare satellites measurements nearby a certain AERONET site and comparing



AERONET measurements nearly synchronized with the satellite overpass time (Sioris et al.
2017). Another reasonable strategy is to average all satellite measurements with a certain
distance of an AERONET location and average all AERONET measurements within a
certain time range (Mishchenko et al., 2010). The results presented in this paper are based
on the second approach as it compares average spatial satellite measurements with average
temporal AERONET measurements. We implemented the Basart et al., (2009) approach in
using a spatial and temporal threshold of 50 km and 30 min for MISR, MODIS, and
AERONET data matching.
We use the Giovanni Multi-sensor Aerosol Products Sampling System MAPSS
(http://giovanni.gsfc.nasa.gov/aerostat/) for the data inter-comparison as aerosols products
are averaged from measurements that are within a radius of ~ 27.5 km from the AERONET
station and within 30 min of each satellite flyover over this location. These data are
represented in the article by MISR / MODIS "matched AERONET data".
"All data" represents AOD products at the selected station. AERONET station 'all data'
are obtained through AEROSOL ROBOTIC NETWORK (AERONET) website
(https://aeronet.gsfc.nasa.gov/). Daily AOD  data with level 2.0 quality was used in the
analysis (Smirnov et al., 2000) . Level 2.0 AOD retrievals are accurate up to 0.02 for mid-
visible wavelengths.
MISR 'all data' is available through MISR website (https://www-
misr.jpl.nasa.gov/getData/accessData/).

**3.  Statistics**
We have used two statistical parameters to compare data retrievals from space-borne and
ground based sensors including:
(1) Correlation coefficient (R),



The correlation coefficient is a parameter to measure data dependence. If the value of R is
close to zero, it indicates weak data agreement. And values close to 1 or -1 indicate that
data retrievals are positively or negatively linearly related (Cheng et al., 2012).

(2) Good Fraction (G- fraction).
The G- fraction indicator uses a data confidence range defined by MISR and MODIS
(Bruegge et al., 1998 and Remer et al., 2005) over the land and ocean that combines
absolute and relative criterion and weights data equally such that small abnormalities will
not affect the inter-comparison statistics (Kahn et al., 2009). In this study, we use MODIS
confidence range which defines data retrieval as "good" if the difference between MODIS
and AERONET is less than
$\Delta \tau = \pm 0.03 \pm 0.05 \tau_{AER}$,   Over ocean,                                              (1)
$\Delta \tau = \pm 0.05 \pm 0.15 \tau_{AER}$,   Over land.                                                (2)

where $\tau_{AER}$ is the optical depth retrieved using AERONET stations. The G-fraction is the
percentage of MODIS data retrievals that satisfies (Equations (1) and (2)) over ocean and
land respectively. Optical depth threshold over land (Equation (1)) is higher than over
ocean (Equation (2)) due to harder data retrievals and high data instability over land.
A good aspect of using data confidence range is excluding small fraction data outliers from
producing inexplicably large influence on comparison statistics by weighting all events
equally.

**4.  Results and discussion**
**4.1 Validating MISR and MODIS AOD retrievals against AERONET observations**
**over the Middle East and North Africa**





Illustrated in Figures 2, 3 and Tables 2, 3 is a regression analysis of MISR and MODIS
Terra AOD products against AERONET AOD over the seven AERONET sites, shown in
Table1, from 2000 – 2015.
The correlation coefficient between MISR and AERONET AOD at region 1 is equal to or
above 0.85 except in Bahrain during DJF and JJA (Figure (2) and Table 2), which could be
attributed to lack of data and the impact of water surface reflectivity over Bahrain. Similar
correlation coefficient values were found in region 2 where MISR-AERONET AOD shows
less error than MODIS (Figures (2, 3) and Table 3). In general, MODIS-AERONET AOD
correlation coefficient is lower than those of MISR at all sites, except Mezaira, where
MISR and MODIS matched AERONET AOD correlation almost match.   The lowest
MODIS-AERONET AOD correlation coefficient was found over Cairo but could be
attributed to the lack of data availability at this location (Figs 3e-h). Low values of
MODIS-AERONET correlation coefficient is also found over Saada, Taman, and Sedee
Boker sites.
Over all AERONET stations, the number of MODIS AERONET matched AOD are 4 to 8
times those of MISR which is expected from the MISR's sampling.
Comparisons show that the difference between MISR and MODIS retrievals at the selected
AERONET sites could be significant as expected from the MODIS Dark Target algorithm
performance over bright land surfaces Kokhanovsky et al. (2007).
High AOD values over regions 1 and 2 measured by both AERONET and satellites'
sensors indicate higher dust activities that peaks during May – Aug during dust storms
season. Higher AOD values recorded during SON over Cairo station could be caused by
seasonal rice straw burning by farmers in Cairo, an environmental phenomena known as
Cairo Black cloud (Marey et al. 2010).  As shown in (Figure (3)), the daily variability in
MODIS measurements is larger than those of MISR at all the three regions. In general,





MODIS tends to underestimate the AOD values on low dust seasons (Figures (2, 3) and
Tables 2, 3).
The MODIS underestimated AOD values are more noticeable over Bahrain. This could be
attributed to large water body surrounding Bahrain, which should affect surface reflectivity.
Moreover, water in the Arabian Gulf has been polluted in recent years (Afnan 2013),
leading to possible changes in watercolour and uncertainties in calculating surface
reflectivity. The patchy land surface or pixel grid contaminated by water body is the
dominant error sources for MODIS aerosol inversion over the land areas (He et al. 2010).
Compared to MODIS, MISR's outperform in retrieving AOD over region 1 including vast
highly reflecting desert areas can be attributed to its multispectral and multi-angular
coverage, which make MISR provide better viewing over a variety of landscapes.
Meanwhile, MISR retrieval also takes into consideration aerosols' particles nonsphericity,
which could have significant effect on its AOD retrievals (von Hoyningen-Huen and Posse
1997). MISR's retrieval did not perform well over Cairo site due to lack of matched points
in most of the seasons (13 in DJF, 5 in MAM & JJA, and 4 in SON during 2000 - 2015).

**4.2 Trends of AOD MISR, MODIS, and AERONET retrievals over the Middle East**
**and North Africa**
Figure 4 shows time series of monthly mean AOD derived from MODIS/Aqua,
MODIS/Terra, MISR and AERONET over a) dust b) biomass and c) mixed dominated
aerosol regions. The satellite AOD trends are calculated from the data collocated with
AERONET observations.
MODIS/ Aqua and MISR AOD at Solar Village have positive trends, while MODIS/ Terra
AOD have negative trends along time series (Fig. 4a). MODIS-Aqua AOD differ from
those of MODIS-Terra. Discrepancy between Aqua and Terra retrievals could be related to





instrument calibration, or the difference in aerosol and cloud conditions from the morning
to the afternoon. Both MODIS Aqua and Terra are underestimating AOD at Solar Village.
MISR AOD trend shows a better agreement with Solar Village AERONET AOD as
compared to MODIS.
Both MODIS/Aqua and MODIS/Terra AOD show a stable trend over time at Mezaria site
with a correlation coefficient of 0.11 and 0.04 respectively. MODIS/Aqua AOD over
Bahrain (not shown in the figure) show, less time trend stability compared to those at Solar
Village with a correlation coefficient 0.63. MODIS/Aqua, MODIS/Terra, and MISR AOD
depicts a positive trend over Cairo, however a 2 years of available AERONET data is not
sufficient for the trend analysis (Fig. 4b). Taman site (Fig. 4c): MODIS/Aqua, MODIS/
Terra, MISR AOD agrees with Taman AERONET on a negative trend indicating data
stability over this site.
Long-range (2000 – 2015) tendency indicates that contradictory AOD trend of Terra and
Aqua is site-dependent and does not necessarily apply everywhere.
AOD difference between Terra and Aqua could be used as another indicator of the long-
range satellites performance. AOD difference (Terra AOD minus Aqua AOD) varies from -
0.01 to 0.19, -0.10 to 0.18, -0.02 to 0.13 over Solar Village, Taman, and Cairo respectively
(Fig. 5). Over the Solar Village, Terra overestimates AOD during 2002-2004 and
underestimates the AOD after 2005. Although Cairo and Taman show similar trend
however over/underestimation amount is not unique for all sites. This is an indication that
Aqua and Terra retrievals disagreement takes place regardless of the region but site
sampling has significant effect on the amount of contradiction.
Statistical comparison between MISR and MODIS/Terra AOD at corresponding
AERONET stations is performed by calculating G-fraction using of $\Delta\tau = \pm0.05 \pm 0.15\tau_{AERO}$
as a confidence interval. Over the region 1, MISR AOD retrievals are more accurate than





MODIS retrievals. MODIS, however, performs better over region 2 sites with high
percentage of the data points falling within the confidence range (Tables 2 and 3). High
light reflections from the desert landscape surrounding region 1 could have an effect on
MODIS retrievals.
Excluding Bahrain and Cairo for low data retrievals the performance of MODIS tends to be
similar over all region with ~ 68 percent of AOD retrievals fall within the
$\Delta\tau = \pm 0.05 \pm 0.15\tau_{AERO}$ confidence range of the AERONET AOD while MISR retrievals
show better performance with ~ 72 percent of the data falling within the same confidence
range. This could be attributed to low number of retrievals available for Bahrain and Cairo
compared to other sites. Vast sea region surrounding Bahrain and complex landscape in
Cairo could also have an impact on retrievals.
**4.3 Evaluating the MISR and MODIS climatology over Middle East and North Africa**
Comparisons between MISR and MODIS AOD at selected AERONET stations over the
2000 – 2015 period are illustrated in Figures 6- 12.
Figure (6a, b) shows histogram of the MISR, MODIS and AERONET AOD at Solar
Village for MISR and MODIS data points collocated with AERONET observations. The
mean, standard deviation, and number of measurements are also presented.
MISR tends to underestimate the frequency of low AOD compared to AERONET but
overestimate the frequency of high AOD. MISR histograms show prominent peaks at 0.55
and 0.75 not seen in AERONET. MISR and AERONET AOD climatology agree well with
one another. MODIS also tends to underestimate the frequency of low AOD events and
overestimate the frequency of high AOD events. High surface reflectance could cause
overestimation in MODIS AOD (Ichoku et al., 2005). Both MISR and MODIS provide a
good representation of the AOD climatology as compared to AERONET at the Solar
Village.  Mezaria station, which is located in an arid region in the UAE, has a similar



climatology to the Solar Village site with dust dominating aerosol. Figure (7a, b) shows
histograms of the MISR, MODIS and AERONET AOD at Mezaria.
Unlike Solar Village, there is a big difference between the number of samples in the
matched data set and full AERONET climatology. For MISR there are 116 matched cases
and for MODIS there are 498 compared to the 1517 for the entire site. This has an impact
on the overall assessment showing significant differences between the matched data and the
full climatology for both MISR and MODIS. First, for the MISR case, the matched
AERONET data have the highest frequency at AOD of 0.3 and 0.35, but the climatology
shows the highest frequency at an AOD of 0.25. Second AOD in the range of 0.3 to 0.45
are oversampled relative to the climatology, and AOD less than 0.3 and greater than 0.5 are
under-sampled with no AOD greater than 0.8. MODIS matched AERONET data show
prominent peaks at 0.3 and 0.4 compared to the climatology that has a single peak at 0.25.
For AOD values between 0.3 and 0.6 MODIS data were found to be under-sampled similar
to MISR AOD.
MISR AOD retrievals matched to AERONET capture the variability in the distribution, but
as in the case of Solar Village the frequency of low AOD events is underestimated and the
frequency of high AOD events is overestimated. However, MISR does capture events with
AOD greater than 1. A similar situation is seen in the MODIS comparison, but MODIS
appears to do a better job capturing the overall shape of the AERONET AOD histogram for
this site.
The Bahrain AERONET site is located in Manama fairly close to the Arabian Gulf, a
location very different from the previous two sites. The site is also located in an urban area
suffers from significant load of anthropogenic aerosols as a consequence of rapid
aluminium industrial development (Farahat 2016). Figure (8a, b) shows histogram of the
MISR, MODIS and Bahrian AERONET measurements with statistical analysis displayed.



The AERONET data matched to MISR show significant peaks at 0.25, 0.35, and 0.5 not
seen in the all data climatology that has a single peak at 0.35. AOD less than 0.25 and
greater than 0.6 are not representative in the matched data set at all. MISR is representing
the peaks at 0.25 and 0.35 in the matched data set but misses the peak at 0.5. The MISR
climatology agrees well with the AERONET all data climatology for all AOD. MODIS on
the other hand shows an extremely large frequency of AOD at 0.1 not represented by
AERONET coupled with an underestimation of AOD greater than 0.3. This could be
attributed to the size of the matching window and MODIS retrievals preferentially coming
from the Arabian Gulf.
SAADA station is located close to some hiking trails at the Agoundis Valley in the Atlas
Mountains about 197 km from the city of Marrakesh.
MISR AOD matched to AERONET agree well with MISR full climatology retrievals over
SAADA station. Both retrievals slightly underestimate SAADA full climatology and over
estimates SAADA matched data retrievals at AOD equal to 0.1 while show good agreement
for AOD greater than 0.1.  MODIS matched to AERONET retrievals overestimate the
frequency of AOD greater than 0.3. While MODIS AOD matched to AERONET captures
climatology at AOD between 0.2 to 0.25, AOD frequency retrievals are under-sampled at
AOD between 0.1 to 0.15 with about 13 % less events than SAADA all data retrievals at
AOD equal to 0.1.
Figure (9a, b) indicates right skewed distribution of SAADA AOD towards small AOD
values with 11.5 % and 30.1 % of AOD > 0.4 as measured by MISR and MODIS
respectively. Taking into consideration MODIS overestimation we conclude that SAADA
site is characterized by small AOD values and this could be related to the land topology
where the station is located.



While MISR is capturing high AOD climatology over SAADA, both MISR and MODIS
are underestimating the frequency of lower AOD events. Nevertheless, MISR captures the
climatology of AOD less than 0.1 missed by MODIS retrievals.
Taman AERONET station is located at the oasis city of Tamanrasset, which lies in
Ahaggar National Park in southern Algeria.
Figure (10 a, b) depicts that Taman AERONET AOD climatology is similar to those at
SAADA and has a high frequency of low AOD events. Both MISR AOD matched to
AERONET and MISR all data do not well capture the frequency of AOD less than 0.1 or
larger than 1 while well describe the climatology for AOD in the range of 0.1 to 1. MODIS
AOD matched data to AERONET correctly describe climatology with slight overestimation
of AOD frequencies between 0.05 – 0.15 while not capturing AOD frequencies greater than
1. MISR and MODIS show similar prominent peaks at 0.1, 0.25, and 0.35, not observed in
Taman AERONET AOD climatology, with more peaks observed by MISR at 0.5, 0.6, and
0.8. Average AOD in SAADA and Taman is ~ 50 percent less than observed at Solar
Village, Mezaria, and Bahrain sites.
Except for AOD greater than 1 where ground observations could be more robust, both
MISR and MODIS retrievals can provide very good climatology matching over Taman site.
Taking into consideration lower number of MISR matching AERONET observations
compared to MODIS ~ 33 and 43 percent over SAADA and Taman respectively, MISR is
outperforming over these two sites which can be attributed to its multiangle viewing
capabilities over complex terrains including mountainous areas (Atlas Mountains).
Cairo is a mega city well known for its high pollution due to traffic and agriculture
activities.
MISR and MODIS matched data correctly capture AOD climatology over Cairo compared
to AERONET as shown in Figure (11a, b). MISR retrievals collocated with AERONET



capture prominent peaks of AERONET AOD at 0.15 – 0.25 and 0.5 with small
underestimation observed at 0.3. MISR 'all data' AOD climatology over Cairo station
agrees better with AERONET AOD climatology vs. collocated dataset with some
oversampling at 0.15. Frequency of high AOD retrievals at 0.7 and 0.8 have not been
captured by MISR matched or all data retrievals. MODIS matched to AERONET AOD are
also able to well present Cairo climatology data with a high overestimation of AOD
frequency between 0.05 - 0.2 and an underestimation of AOD larger than 0.4.
The complex landscape and local emissions in Cairo could impose major challenges in
MODIS AOD retrievals. Moreover, Cairo is one of the most densely populated cities in the
world that hosts major commercial and industrial centers in North Africa. Cairo also has
complicated aerosols structure developed by long range transported dust in the spring,
biomass burning in the fall, strong traffic and industrial emissions (Marey et al., 2010).
Over Cairo station, MODIS correctly represents ground observations for AOD between 0.2
- 0.4 while MISR all data better represents AOD climatology for AOD greater than 0.4.
There is not enough collocated MISR-AERONET AOD to evaluate MISR 'matched AOD'
climatology.
MISR, MODIS climatology at SEDEE Boker are illustrated in Figures (12a, b).
MISR 'matched' AOD frequency show significant underestimation for AOD less than 0.2
and an overestimation between 0.2 – 0.4 compared with AERONET retrievals. MISR
correctly captures the climatology for AOD events greater than 0.4. MISR 'matched' and
'all data' retrievals peaks at 0.25 and 0.2 respectively producing high frequency of AOD
oversampling compared to AERONET. MISR data retrievals do not capture the
climatology for AOD less than 0.1 over this site coincident with what was previously
observed over other sites. MODIS matched AERONET data underestimates frequency of
AOD less than 0.2 while overestimates the frequencies between 0.2 - 0.6, and well match



frequencies of higher AOD events larger than 0.6. MODIS retrievals are characterized by
two prominent peaks at 0.1 and 0.25 that are not found in the AERONET matched data.
At Sedee, MISR and MODIS retrievals are better in matching frequency of high AOD
retrievals (greater than 0.4) than the frequency of low AOD. This could be an effect of
possible long-range transport to Sedee Boker site (Farahat et al. 2016) along with complex
mixtures of dust, pollution, smoke, and sea salt that could result in uncertainties in MISR
and MODIS aerosol model selection.
In the summary, MISR tends to underestimate AOD > 0.4 over Solar Village, Mezaria,
Bahrain, and Cairo while agrees with AERONET over SAADA, Taman and Sedee Boker
at all ranges of AOD. This could be expounded by insufficient particle absorption in MISR
V22 algorithm (Kahn et al., 2005). Spherical particle absorption is produced by externally
mixing small black carbon particles.
Percentage of MISR, MODIS, and AERONET AOD greater than 0.4 recorded is shown in
Table 4. Over Solar Village, both MISR and MODIS well capture high AOD greater than
0.4 with very good agreement with the ground observations. Over Mezaria, both MISR and
MODIS are over estimating the percentage of AOD greater than 0.4 by about 15.5 and 10.5
percent respectively. MISR all data agrees well with AERONET all data in representing
high AOD over Bahrain while MODIS shows significant under-representation of those
events by about 15 percent, less than reported by Bahrain AERONET station. At SAADA,
MISR AOD agrees with AERONET in showing low percentage of AOD greater than 0.4,
while MODIS retrievals overestimate percentage by about 24 percent. MISR AOD over
Taman AERONET station shows very good agreement, while MODIS is slightly
overestimating AOD. Among all seven sites considered in this study, Sedee Boker shows
lowest occurrence of AOD greater than 0.4, which is confirmed by both MISR and MODIS





retrievals. Cairo AERONET records the highest frequency of AOD > 0.4, however this is
largely underestimated by both MISR and MODIS retrievals.
It can concluded from the previous discussion that atmosphere around SAADA, Taman,
and Sedee Boker sites is relatively clean and aerosol loads are small compared to Solar
Village, Mezaria, Bahrain, and Cairo, however this could be affected by the location where
AERONET station is installed for example SAADA and Taman stations are installed in a
remote mountainous region away from urbanization while Cairo station is installed in the
middle of large residential region with significant local emissions.

**Conclusion**
The performance of MODIS, MISR retrievals with corresponding AERONET
measurements over different geographic locations in the Middle East and North Africa was
investigated during 2000 – 2015.
Long-range observations show dissimilar AOD trends between MODIS/Aqua,
MODIS/Terra, MISR and AERONET measurements. MODIS/Aqua matched AERONET
retrievals show stable trend over all sites while, MODIS/Terra matched AERONET
retrievals show significant downward trend indicating possible changes in the sensor
performance.
MISR matched AERONET AOD data depict high correlation compared to
AERONET indicating good agreement with ground observations with about 72 percent of
AOD retrievals fall within the expected confidence range.
Consistency of MODIS and AERONET AOD vary based on the season, study area,
and dominant aerosols type with about 68 percent of the retrieved AOD values fall within
expected confidence range with the lowest performance over mixed particles regions.



Comparing satellites' AOD retrievals with corresponding AERONET
measurements show that space-borne data retrievals accuracy can be affected by landscape,
topology, and AOD range at which data is retrieved.
Few AERONET sites are verified where MISR and MODIS retrievals agree well
with ground observations, while other sites only MISR or MODIS could correctly describe
the climatology.
The AOD range at which MISR or MODIS could correctly describe ground
observation is also investigated over different AERONET sites. Over Solar Village both
MISR and MODIS tend to underestimate the frequency of low AOD and overestimate the
frequency of high AOD compared to AERONET with MISR histograms show prominent
peaks at 0.55 and 0.75 not shown in AERONET. MISR can capture the frequency of AOD
greater than 1 mostly missed by MODIS. Both MISR and MODIS are found to provide
good representation of the AOD climatology over the Solar Village site.
Similar to Solar Village, MISR underestimates frequency of lower AOD and
overestimate frequencies of high AOD over Mezaria. MISR is able to correctly capture the
frequency of AOD greater than 1, while MODIS retrievals are found to better represent the
overall climatology. This is due to low number of MISR – matched AERONET retrievals
compared to MODIS over this site. Prominent peaks at 0.3 and 0.4 were observed in
MODIS matched Mezaria retrievals compared to the climatology, which has a single peak
at 0.25.
Large water body surrounding Bahrain makes MODIS data preferentially originate
from the Arabian Gulf which produces an extremely large frequency of AOD at 0.1 not
observed in AERONET measurements paired with an underestimation of AOD greater than
0.3. Meanwhile, MISR retrievals agree well with AOD climatology over Bahrain.





MISR AOD retrievals slightly underestimate SAADA climatology while show good
agreement for AOD greater than 0.1.  MODIS retrievals underestimate the frequency of
AOD retrievals between 0.1 to 0.15, match climatology at AOD between 0.2 to 0.25, and
overestimate the frequency of AOD greater than 0.3.  SAADA site is characterized by
small frequency of low AOD values and this could be related to the landscape nature
surrounding Saada station. MISR is found to be outperforming over Saada and Taman
stations which can be attributed to its viewing multispectral and multiangular capabilities
over mountainous regions.

MISR retrievals well capture prominent peaks of AERONET data at 0.15 to 0.25

and 0.5 with small underestimation observed at 0.3 over Cairo. It is recommended to use
MISR all data rather than matched data only over Cairo as it is found to do a better job in
describing the climatology over this station. MODIS data retrievals are also able to well
present Cairo climatology with a high overestimation of AOD frequency between 0.05 to
0.2 and an underestimation of AOD larger than 0.4. While both MISR and MODIS well
describe climatology over Cairo station, MODIS can correctly represent ground
observations between 0.2 to 0.4.
Over Sedee Boker both MISR and MODIS retrievals well describe the climatology
however they are more successful in matching frequency of high AOD greater than 0.4.
Based on analysing frequency of AOD greater than 0.4, it was found that Saada, Taman,
and Sedee Boker are having better air quality compared to other sites while Cairo was
found to be the most polluted site.
Results presented in this study are important in providing a guideline for satellites retrievals
end users on which sensor could provide reliable data over certain geographic location and
AOD range.





Adjacent geographic location and local climate among sites does not always

guarantee that same sensor will provide consistent retrievals over all sites. For example,
Solar Village, and Bahrain AERONET are surrounded by large desert regions in the and
sharing almost similar climatic conditions, but MODIS is found to be more successful in
describing climatology over Solar Village than over Bahrain and this could be attributed to
different factors related to surface reflection, cloud coverage, and the large water body
surrounding Bahrain. Thus in order to decrease data uncertainty, it is important to
determine which sensor provides best retrieval over certain geographic location and AOD
range.

**Acknowledgements**
The author would like to acknowledge the support provided by the Deanship of Scientific
Research (DSR) at the King Fahd University of Petroleum and Minerals (KFUPM) for
funding this work through project # IN161053. Portions of this work were performed at the
Jet Propulsion Laboratory (JPL), California Institute of Technology, under a contract with
the National Aeronautics and Space Administration. The author would like to thank
Michael Garay (MJG) and Olga Kalashnikova (OVK) (JPL) for their suggestion of
investigating satellites – AERONET matched data climatology, and discussion during the
data analysis. The author would also like to thank Hesham El-Askary (Chapman
University) for providing recommendation about AERONET data over North Africa and
the Middle East as well as reviewing the English in the manuscript. We thank the MISR
project for providing facilities, and supporting contributions of MJG and OVK. Finally, we
thank the reviewers for suggestions, which improved the manuscript.
**Author Contributions**: Ashraf Farahat analysed the data, performed the statistical analysis
and wrote the manuscript.
**Conflicts of Interest**: The authors declare no conflict of interest.



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



**Tables' caption**
Table 1. Geographic location of the AERONET sites used in this study
Table 2. Statistics for dust sites, R: correlation coefficient, RMSE: Root Mean Square
deviation; G-fraction: good fraction; N: number of observations
Table 3. Statistics for biomass and mixed sites, parameters as in Table 3. Caption.
Table 4. MISR coverage for six days of major dust activity over the Arabian Peninsula
during March 2009.





**Figures caption**
Figure 1. Location of the AERONET stations over North Africa and the Middle East. The
numbers on the map indicates the site location as 1: Saada, 2: Tamanrasset_INM, 3: Cairo,
4: Sede Boker, 5: Solar Village, 6: Mezaira, 7: Bahrain.
Figure 2. Scatter plot of MISR AOD versus AERONET AOD based on seasons and
aerosols categorization.
Figure 3. Scatter plot of MODIS AOD versus AERONET AOD based on seasons and
aerosols categorization.
Figure 4. Time series of monthly mean AOD derived from MODIS/Aqua, MODIS/Terra,
MISR and AERONET over a) dust b) biomass and c) mixed dominated aerosol regions.
Figure 5. Long range AOD difference for MODIS/Terra and MODIS/Aqua over the dust,
biomass and mixed sites.
Figure 6. Histogram of the MISR, MODIS and Solar Village AERONET measurements a)
MISR b) MODIS data retrievals.
Figure 7. Histogram of the MISR, MODIS and Mezaria AERONET measurements a)
MISR b) MODIS data retrievals.
Figure 8. Histogram of the MISR, MODIS and Bahrain AERONET measurements a) MISR
b) MODIS data retrievals.
Figure 9. Histogram of the MISR, MODIS and SAADA AERONET measurements a)
MISR b) MODIS data retrievals.
Figure 10. Histogram of the MISR, MODIS and Taman AERONET measurements a)
MISR b) MODIS data retrievals.





Figure 11. Histogram of the MISR, MODIS and SEDEE Boker AERONET measurements
a) MISR b) MODIS data retrievals.
Figure 12. Histogram of the MISR, MODIS and Cairo AERONET measurements a) MISR
b) MODIS data retrievals.


























Table 1.

| Location name | Lon./Lat. | Measurement period |
|---|---|---|
| Solar Village | 24.907$^o$ N/46.397$^o$ E | 2000-2015 |
| Mezaria | 23.105$^o$ N/53.755$^o$ E | 2004-2015 |
| Bahrain | 26.208$^o$ N/50.609$^o$ E | 2000-2006 |
| Saada | 31.626$^o$ N/8.156$^o$ W | 2003-2015 |
| Taman | 22.790$^o$ N/5.530$^o$ E | 2000-2015 |
| Cairo | 30.081$^o$ N/31.290$^o$ E | 2005 -2007 |
| Sede Boker | 30.855$^o$ N/34.782 $^o$ E | 2000-2015 |


















Table 2.

| AERONET Site | Sensor | Season | Mean Value | | N | R | Gfraction (%) |
|---|---|---|---|---|---|---|---|
| | | | AERONET | Satellite | | | |
| Solar Village | MISR | DJF | 0.31±0.22 | 0.38±0.20 | 338 | 0.94 | 60.05 |
| | | MAM | 0.39±0.27 | 0.45±0.23 | 89 | 0.94 | 65.16 |
| | | JJA | 0.39±0.18 | 0.45±0.17 | 141 | 0.90 | 70.21 |
| | | SON | 0.27±0.16 | 0.35±0.14 | 3 | 0.99 | 33.33 |
| | MODIS Terra | DJF | 0.27±0.19 | 0.33±0.17 | 1500 | 0.48 | 51.80 |
| | | MAM | 0.36±0.24 | 0.26±0.17 | 389 | 0.68 | 90.23 |
| | | JJA | 0.34±0.17 | 0.42±0.19 | 429 | 0.41 | 54.31 |
| | | SON | 0.22±0.10 | 0.36±0.12 | 471 | 0.51 | 28.87 |
| Mezaria | MISR | DJF | 0.33±0.15 | 0.40±0.17 | 60 | 0.89 | 75.00 |
| | | MAM | 0.32±0.19 | 0.41±0.22 | 13 | 0.90 | 69.23 |
| | | JJA | 0.42±0.13 | 0.47±0.17 | 21 | 0.85 | 80.95 |
| | | SON | 0.29±0.07 | 0.36±0.07 | 22 | 0.87 | 77.27 |
| | MODIS Terra | DJF | 0.32±0.15 | 0.35±0.19 | 198 | 0.86 | 74.74 |
| | | MAM | 0.44±0.33 | 0.45±0.27 | 115 | 0.92 | 78.07 |
| | | JJA | 0.39±0.14 | 0.43±0.20 | 89 | 0.81 | 71.91 |
| | | SON | 0.28±0.13 | 0.30±0.16 | 97 | 0.87 | 77.31 |
| Bahrain | MISR | DJF | 0.37±0.11 | 0.31±0.10 | 17 | 0.73 | 100 |
| | | MAM | 0.31±0.11 | 0.28±0.14 | 3 | 0.89 | 100 |
| | | JJA | 0.40±0.09 | 0.36±0.09 | 8 | 0.69 | 100 |
| | | SON | 0.40±0.09 | 0.30±0.05 | 4 | 0.98 | 100 |
| | MODIS Terra | DJF | 0.42±0.29 | 0.20±0.19 | 121 | 0.41 | 93.38 |
| | | MAM | 0.50±0.28 | 0.13±0.15 | 25 | 0.26 | 96.00 |
| | | JJA | 0.55±0.26 | 0.31±0.27 | 42 | 0.50 | 88.09 |
| | | SON | 0.35±0.14 | 0.21±0.12 | 29 | 0.32 | 93.10 |





Table 3.

| AERONET Site | Method | Season | Mean Value | | N | R | Gfraction (%) |
|---|---|---|---|---|---|---|---|
| | | | AERONET | Satellite | | | |
| SAADA | MISR | DJF | 0.24±0.16 | 0.22±0.15 | 149 | 0.93 | 97.29 |
| | | MAM | 0.21±0.13 | 0.19±0.11 | 53 | 0.89 | 96.15 |
| | | JJA | 0.29±0.14 | 0.27±0.15 | 80 | 0.93 | 97.46 |
| | | SON | 0.19±0.15 | 0.19±0.12 | 60 | 0.94 | 98.30 |
| | MODIS Terra | DJF | 0.23±0.16 | 0.32±0.21 | 550 | 0.57 | 57.81 |
| | | MAM | 0.24±0.18 | 0.39±0.23 | 90 | 0.43 | 44.44 |
| | | JJA | 0.30±0.17 | 0.45±0.18 | 201 | 0.40 | 45.27 |
| | | SON | 0.19±0.13 | 0.22±0.14 | 162 | 0.71 | 72.39 |
| Taman | MISR | DJF | 0.19±0.23 | 0.24±0.19 | 135 | 0.92 | 70.89 |
| | | MAM | 0.29±0.22 | 0.35±0.24 | 24 | 0.97 | 82.60 |
| | | JJA | 0.35±0.30 | 0.39±0.19 | 36 | 0.85 | 71.42 |
| | | SON | 0.19±0.15 | 0.19±0.12 | 60 | 0.94 | 98.30 |
| | MODIS Terra | DJF | 0.19±0.22 | 0.18±0.16 | 319 | 0.67 | 81.81 |
| | | MAM | 0.24±0.19 | 0.22±0.17 | 67 | 0.55 | 83.58 |
| | | JJA | 0.37±0.32 | 0.29±0.20 | 69 | 0.69 | 84.05 |
| | | SON | 0.14±0.14 | 0.13±0.10 | 117 | 0.54 | 84.61 |
| Cairo | MISR | DJF | 0.33±0.20 | 0.28±0.11 | 13 | 0.94 | 100 |
| | | MAM | 0.22±0.06 | 0.24±0.08 | 5 | 0.99 | 100 |
| | | JJA | 0.43±0.23 | 0.34±0.11 | 5 | 0.99 | 100 |
| | | SON | 0.38±0.21 | 0.29±0.12 | 4 | 0.97 | 100 |
| | MODIS Terra | DJF | 0.33±0.16 | 0.20±0.11 | 158 | 0.30 | 95.56 |
| | | MAM | 0.32±0.16 | 0.12±0.08 | 39 | 0.25 | 100 |
| | | JJA | 0.35±0.14 | 0.28±0.07 | 58 | 0.17 | 94.82 |
| | | SON | 0.38±0.19 | 0.20±0.09 | 29 | 0.07 | 93.82 |
| | | DJF | 0.14±0.06 | 0.21±0.07 | 23 | 0.87 | 40.90 |





| | | | | | | | |
|---|---|---|---|---|---|---|---|
| | | MAM | 0.14±0.05 | 0.24±0.09 | 13 | 0.68 | 33.33 |
| | MISR | JJA | 0.16±0.05 | 0.24±0.06 | 163 | 0.85 | 33.33 |
| | | SON | 0.15±0.07 | 0.23±0.06 | 72 | 0.89 | 33.80 |
| SEDEE_BOKER | | DJF | 0.16±0.12 | 0.23±0.14 | 1312 | 0.36 | 53.50 |
| | | MAM | 0.21±0.18 | 0.24±0.19 | 338 | 0.34 | 65.68 |
| | MODIS | JJA | 0.16±0.09 | 0.33±0.13 | 392 | 0.27 | 17.34 |
| | Terra | SON | 0.16±0.09 | 0.23±0.12 | 477 | 0.46 | 58.49 |
























Table 4.

|  | AERONET | | MISR | | MODIS | |
|---|---|---|---|---|---|---|
|  |  | AOD | | AOD | | AOD |
|  | N | % > 0.4 | N | % > 0.4 | N | % > 0.4 |
| Solar Village | 3978 | 28.7 | 684 | 32.8 | 2789 | 30.1 |
| Mezaria | 1650 | 30.2 | 547 | 45.7 | 498 | 40.7 |
| Bahrain | 1117 | 33.3 | 676 | 35.7 | 217 | 18.4 |
| SAADA | 3184 | 10.8 | 667 | 11.5 | 1004 | 34.6 |
| Taman | 1863 | 17.9 | 845 | 22.6 | 572 | 9.4 |
| Cairo | 269 | 53.5 | 620 | 17.7 | 284 | 4.2 |
| SEDEE | 5722 | 4.8 | 675 | 9 | 2519 | 12.8 |






























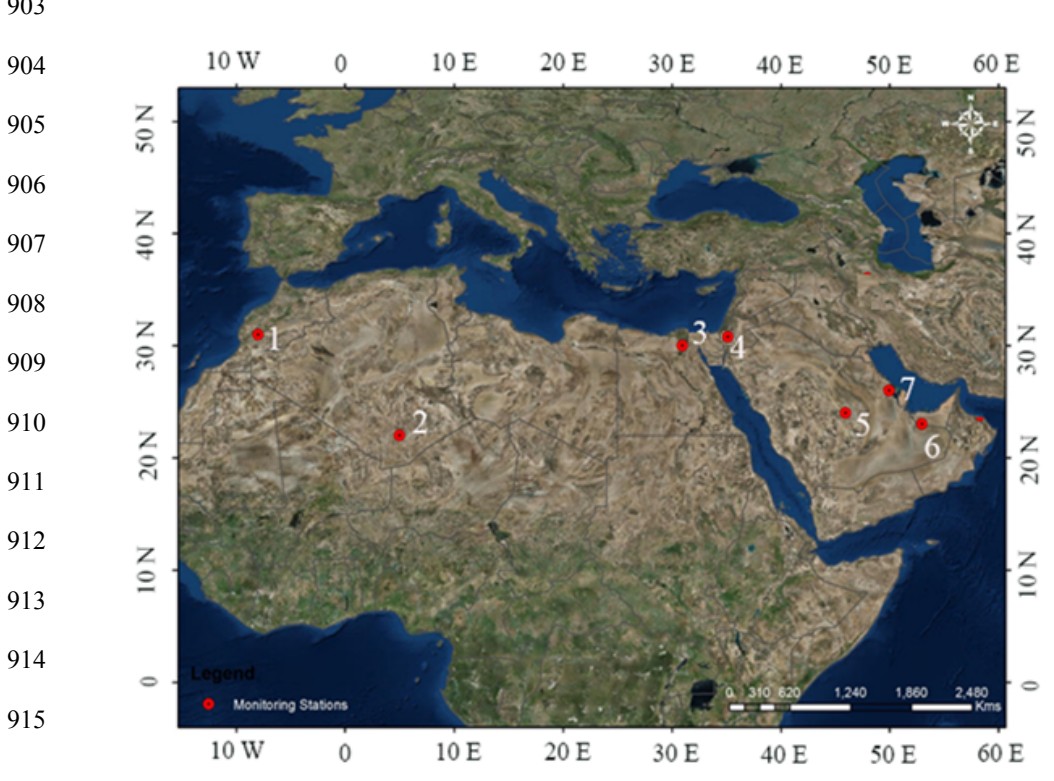

Figure 1.















Figure 2.



Figure 3.





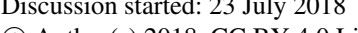




Figure 4.

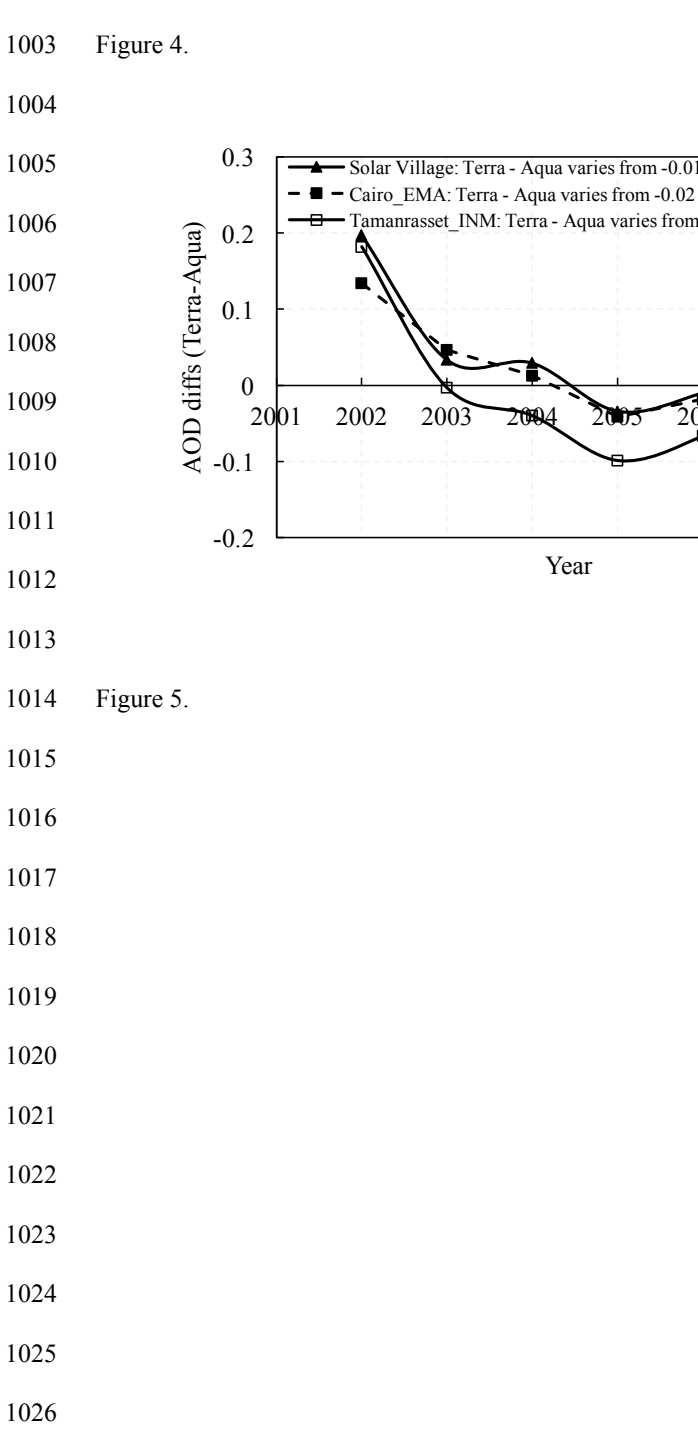

Figure 5.

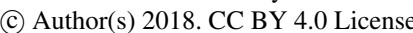




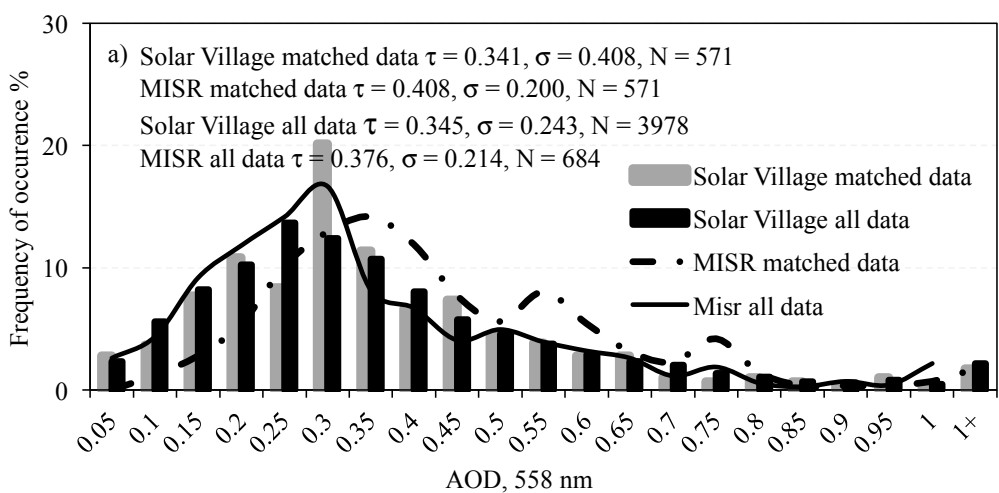


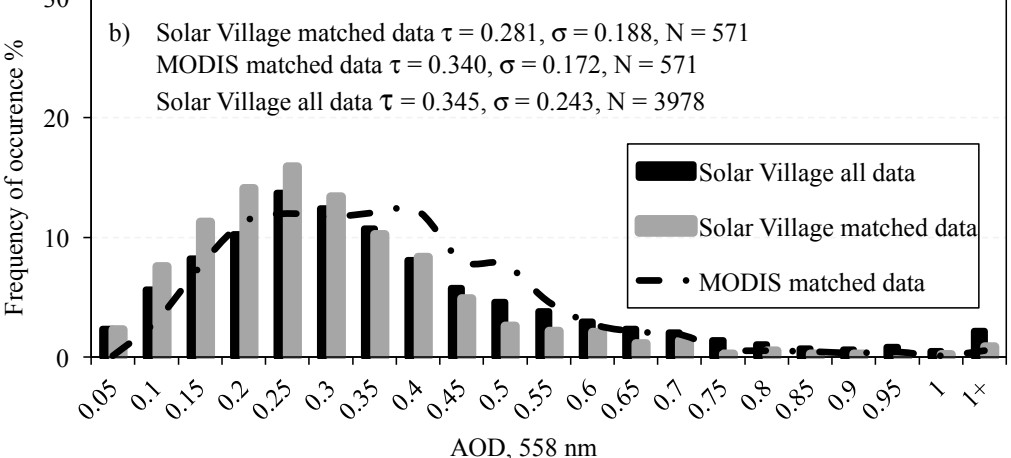


Figure 6.





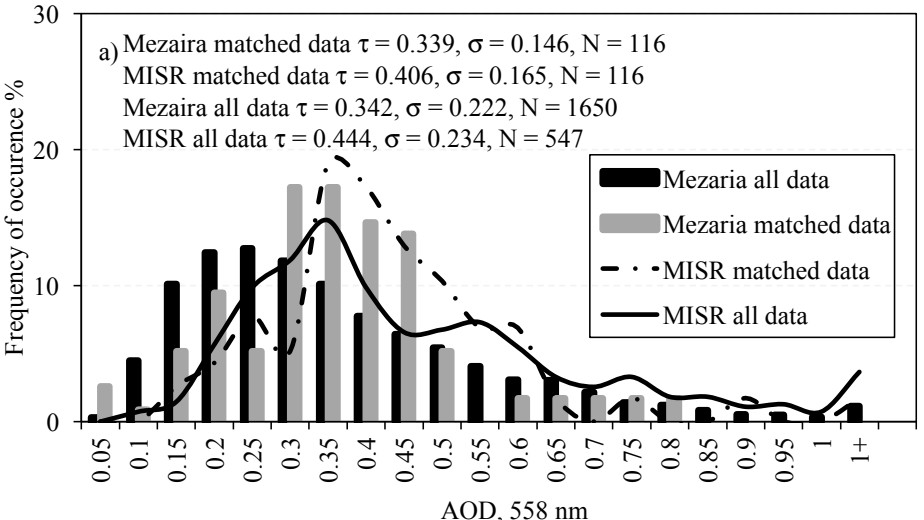

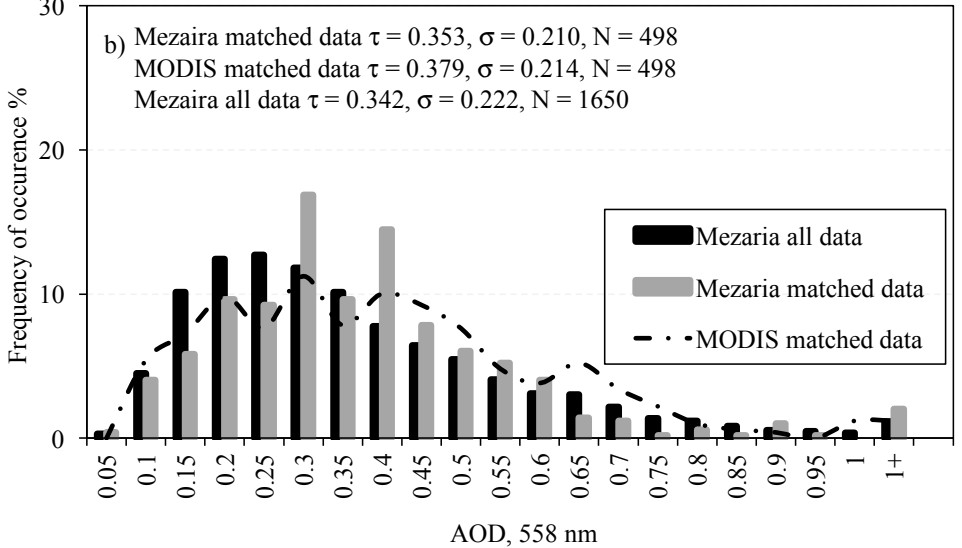




Figure 7.






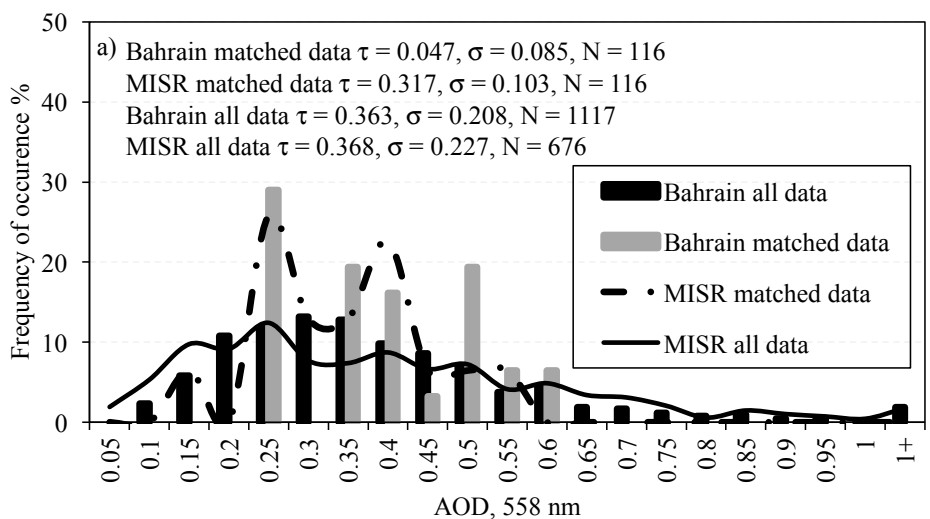

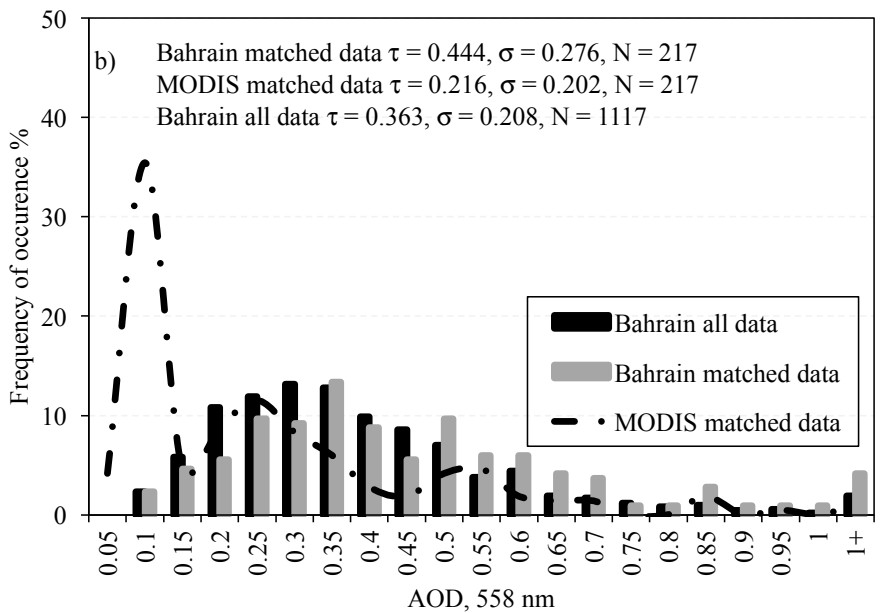

Figure 8.




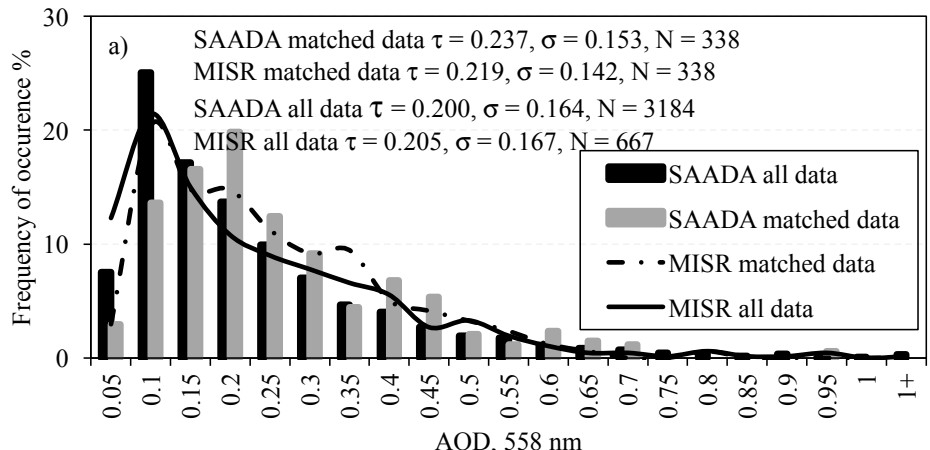


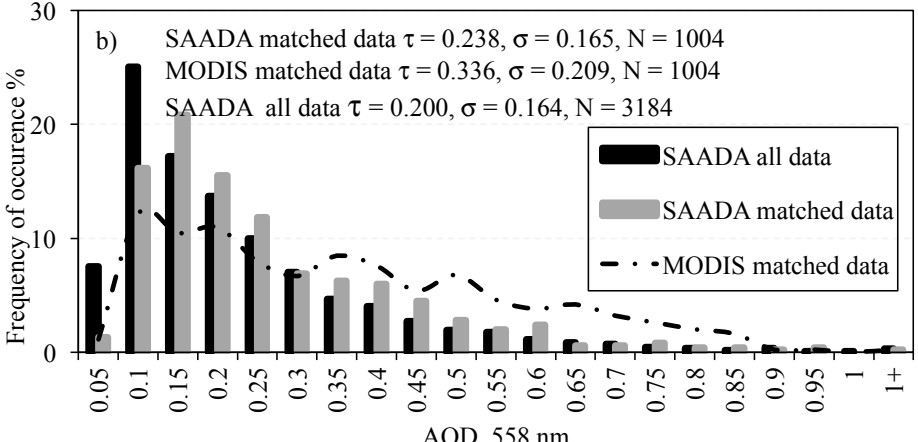



Figure 9.








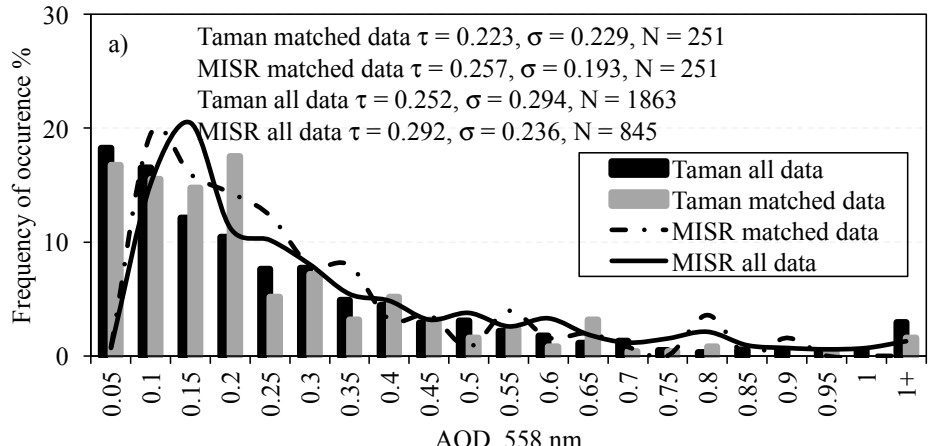



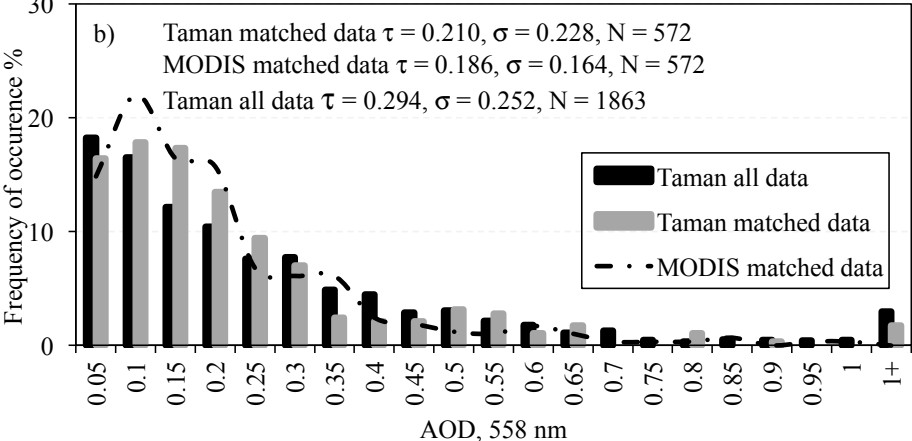




Figure 10.





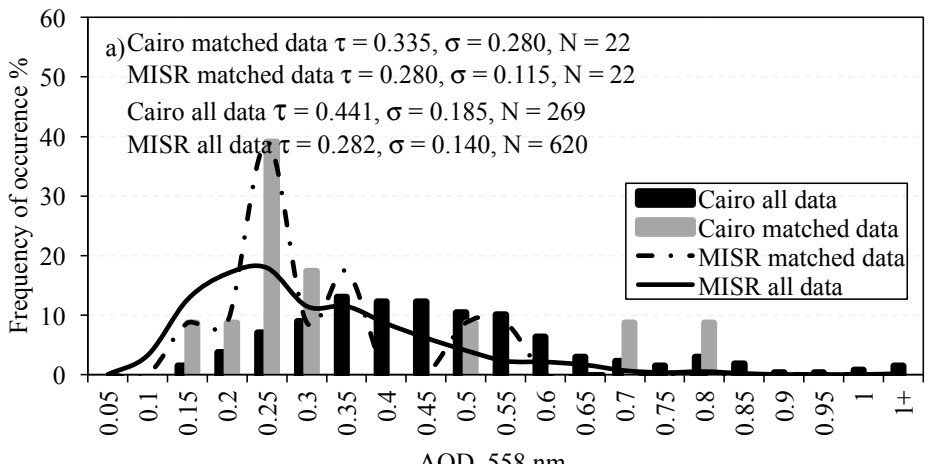



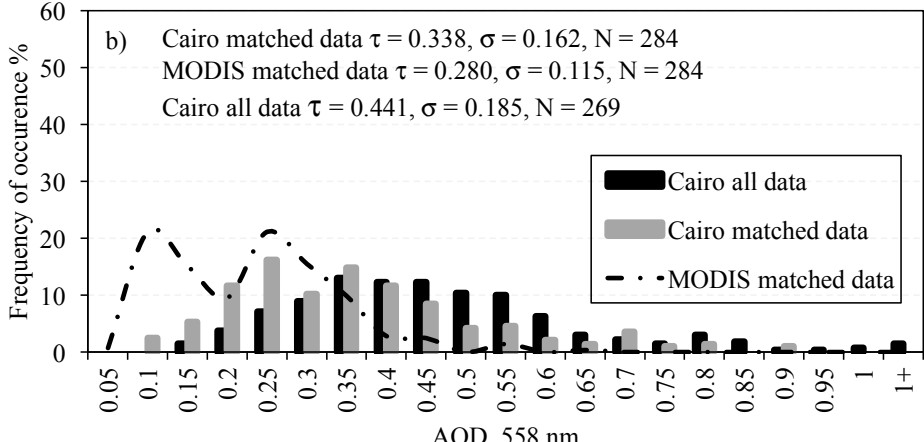





Figure 11.






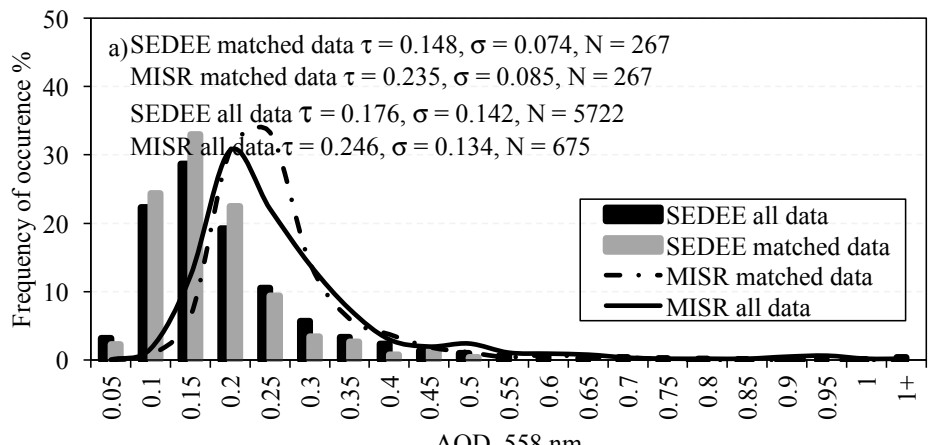



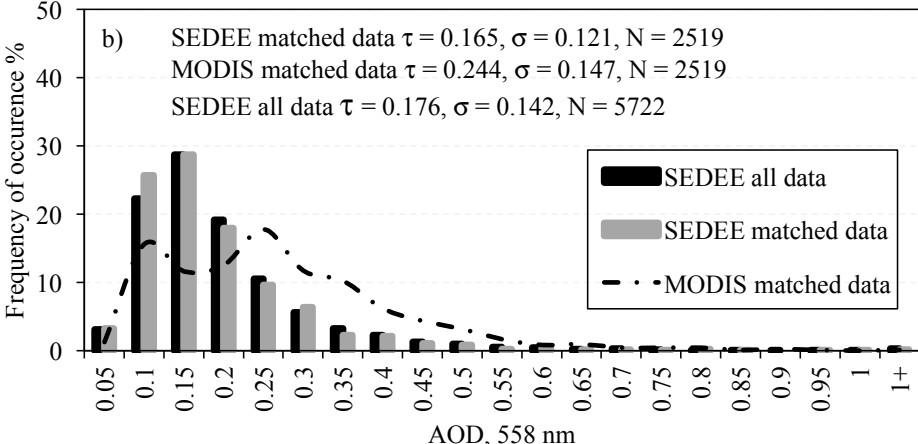



Figure 12.