# Peer review of "Comparative Analysis of MODIS, MISR and AERONET Climatology 1 over the Middle East and North Africa 2 3 Ashraf Farahat 4 Department of Physics, King Fahd University of Petroleum and Minerals, Dhahran 31261, 5 Saudi Arabia; E-Mails: farahata@kfupm.edu.sa 6 7 \*Author to whom correspondence should be addressed; E-Mail: farahata@kfupm.edu.sa. 8 Tel: (321) 541-7088 9 10 Abstract: 11 Comparative analys"

_Annales Geophysicae, 2018_

## Short Comment (SC1) · 24 Jul 2018

A. M. Sayer

andrew.sayer@nasa.gov

This is not a review, only a short comment about the data versions used in this study. I work on or with the teams responsible for these data products (MODIS, MISR, AERONET). All three recently (within the last year) released new data versions, and older versions should be considered obsolete. It is not clear which versions were used here in some cases, and I'd strongly encourage use of the latest versions, so the paper is not outdated before it is published. These new versions quantitatively change the spatial patterns and magnitude of the AOD, and so directly influence all the results presented in this study.

The author is using version 22 MISR data. The latest version is version 23. Version

23 has numerous algorithm/calibration improvements. The MISR part of this study is therefore out of date.

So far as I could tell, the author does not state which version of MODIS data is being used. The latest is Collection 6.1. Collection 6.1 also has a lot of updates over the previous Collection 6. The data set description the author provides in the paper is for the Dark Target algorithm. However, Dark Target does not provide data over the bright surfaces analysed in this study. I therefore think that it is likely that the author is instead using data derived using the Deep Blue algorithm. Deep Blue and Dark Target are distinct algorithms; I encourage the author to double-check and update data version and descriptions as necessary.

I also did not see which AERONET data version the author is using. The latest is version 3. This also has numerous improvements over version 2. Note that "version" and "level" mean different things in AERONET: the right data version to use is version 3, level 2 data.

This analysis includes time series and trend calculations. The latest MODIS and MISR versions include updated sensor calibration, which is important because it corrects artificial calibration drifts which can cause apparent false trends in AOD. The impact on the trends is hard to state because it depends on many factors (it isn't a simple linear mapping of calibration drift into AOD drift). It is therefore likely that, if the latest versions of the data are used, different results will be obtained for the trend calculations.

Since it looks like the author got the data through the Giovanni visualisation interface. It is generally best to go to the official data centers to make sure you get the latest data versions and accompanying information. These links are:

MODIS: https://ladsweb.modaps.eosdis.nasa.gov/

MISR: https://eosweb.larc.nasa.gov/

AERONET: https://aeronet.gsfc.nasa.gov

All these data sets are freely available. I am not sure when/if these updates will be reflected in Giovanni, as that website is not maintained by the algorithm teams.

I'd be happy to help if the authors has questions about obtaining and using these latest data versions.

---

## Referee Comment (RC1) · Anonymous Referee #1 · 24 Aug 2018

The author presents an analysis of collocated MISR and MODIS satellite AOD retrieval data over seven AERONET sites for a 16-year period. These sites are located in or next to the desert regions of North Africa and the Middle East, hence many (but not all) of these sites have aerosol conditions dominated by desert dust. It is a straightforward study, which seeks to analyse the behaviour of the various AOD products with respect to each other.

The histograms in figures 6-12 are the most useful depiction here of the collocated datasets, displaying the distributions of AOD values over the various AERONET sites as retrieved by AERONET and the satellite products. It is interesting that the MISR AOD retrieval does not appear to capture the very low AODs observed by AERONET. However the trend analysis provides a rather weak discussion and conclusion, only

hinting at significant values for the Solar Village site with AERONET and MISR, as far as I can see from the figure.

I noticed the short comment by Andrew Sayer (I usually try to avoid reading other reviews in discussion journals, but as a comment on data versions this seemed to be a particularly relevant point), and I agree that it is vital that the most up-to-date data versions are used for all three of the datasets. If the current versions are not used then the analysis in this paper is of only minimal historical interest. Therefore please make sure that you are using the new Version 3 AERONET products, for example. I do not know how much difference to the results re-performing the analysis will cause, but presumably there will be differences in almost all of the figures and tables.

Please also clarify whether you are using the Dark Target (DT) and/or the Deep Blue (DB) AOD retrievals, since these use very different retrieval methods, and it is a vital distinction to make. Presumably the MODIS AODs over central desert sites such as Solar Village or Tamanrasset would be from the Deep Blue algorithm, while coastal sites such as Bahrain would have a greater prevalence of DT retrievals. It would per-haps make more sense to discriminate the MODIS AODs further, between retrievals using the DT and the DB algorithms. A possible question might be whether the DB or the DT algorithm performs better in the vicinity of Bahrain or other such sites on the desert margins?

Throughout the manuscript there are language issues which should be corrected.

Specific Comments

p.2, lines 36-37: why is this in italics?

Section 2.2: if MODIS Deep Blue retrievals are used (and they should be), please also describe them here.

p.14, line 330: do you know what these peaks indicate? On brief speculation I might imagine that the first peak is indicative of industrial aerosol and the second peak might

be indicative of dust. Ångström coefficient values may give some evidence as to what these might be.

p.14, lines 337-338: if the MODIS retrievals are preferentially coming from the Gulf, does that mean that the great majority of the retrievals over Bahrain are from DT?

p.14, line 253: 'topology'. I think you mean 'topography'?

Figure 4(b): I thought Cairo only had AERONET data from 2005-2007?

---

## Referee Comment (RC2) · Anonymous Referee #2 · 5 Sep 2018

The author of this manuscript has done quite interesting work, well analyzed "Comparative Analysis of MODIS, MISR 1 and AERONET Climatology over the Middle East and North Africa". In general the manuscript is interesting and well written. The results have been presented and discussed well and thoroughly. In my opinion, the topic discussed in this paper is suitable for publication. Overall I recommend acceptance of this paper for publication with minor revisions. Please see the specific comments below.

Minor comments:

Line 11: please insert comma after MISR Line 15: please check the grammar, i.e. MODIS/terra AOD indicates instead of indicate Line 33: please use like this "that has major effects on human activities in the Arabian" Line 42-43: please make it clear to the reader. Line 121: please rephrase the sentence. Line 136-137: please rephrase the

sentence. Line 142: please mention the name of satellite Line 147-149: please revise the sentence. Line 158: The authors have mentioned that they have used second approach in this study. Why did the authors not use the first approach? Line 176: The authors have used only two statistics parameters to validate the satellite data. It is suggested to use more parameters for the validation. It is also observed that authors have not mention the value of statistical parameters in the figures. Line 196: please correct number of equations in the text.

Table 2: Caption of table should be precise and general and table value should match according to the caption e.g RMSE is mentioned in the caption but not presented in the table, G-fraction and Gfraction should same in the text. Table 3: Like statistics for biomass and mixed, parameter as in table 2 (but you mentioned parameter as table 3) Second column of each table should be same if they belongs to same category. It will confuse the reader, like in table 2, you used ‘sensor’ but in table 3 you changed sensor to ‘method’ but they are the same indeed. It will confuse the reader

Table 4: Caption of table 4 is again confusing MISR coverage but in the body of table MODIS, MISR and AERONET are all showing their coverage

FIGURE 1: Check the grammar of caption of figure1 e.g. “The numbers on the map indicate, not indicates” What is the source of this fig? Please combine figure 2 and 3 because they are the same actually just with different satellite data

---

## Author Comment (AC2) · 19 Sep 2018

A. Farahat farahata@kfupm.edu.sa

Response to Anonymous Referee # 1 Dear Referee, Thank you very much for your feedback about our article. We greatly appreciate the comments. We have addressed all your comments and we have revised the manuscript accordingly. For your consideration, we have included a copy of the revised article with track changes.

Please find below our response to your comments. Regards,

Ashraf Farahat

Comments from Referees The author presents an analysis of collocated MISR and MODIS satellite AOD retrieval data over seven AERONET sites for a 16-year period. These sites are located in or next to the desert regions of North Africa and the Middle

[Figure]

East, hence many (but not all) of these sites have aerosol conditions dominated by desert dust. It is a straightforward study, which seeks to analyse the behaviour of the various AOD products with respect to each other Author's response We agree with the reviewer's comment that not all sites covered by this study are dominated by desert dust. It has been reported by Farahat et al. 2016 (Farahat, A., El-Askary, H., Adetokunbo, P., and Fuad, A.-T.: Analysis of aerosol absorption properties and transport over North Africa and the Middle East using AERONET data, Ann. Geophys., 34, 1031-1044) that dominated aerosols over these sites are seasonal and location dependent. They also depend on local pollution and aerosols transport.

Comments from Referees The histograms in figures 6-12 are the most useful depiction here of the collocated datasets, displaying the distributions of AOD values over the various AERONET sites as retrieved by AERONET and the satellite products. It is interesting that the MISR AOD retrieval does not appear to capture the very low AODs observed by AERONET. However the trend analysis provides a rather weak discussion and conclusion, only hinting at significant values for the Solar Village site with AERONET and MISR, as far as I can see from the figure. Author's response It is important to mention that the goal of this study is to assess the consistency in aerosol trends between spaceborne sensors and AERONET data. The study tried to investigate which satellite data can better describe ground-based measurements over certain geographic locations in the Middle East and North Africa. Our analysis mainly focused on how data availability, topography, and water areas can affect satellite's measurements from one region to another. Aerosols categorization and sources are not the major focus of this study. Having this in mind, following the reviewer's comment we have revised the discussion as below: The following paragraphs have been added to section 4.2 p.11, lines 286-288 new paragraph added Trends of aerosol loading from 2000 to 2005 are analysed by plotting fitting lines of monthly mean AOD retrievals by MISR and MODIS/Terra and Aqua. The AOD retrieved by different instrument shows different trends. p.11, lines 290-293 new paragraph added Terra depicts a negative correlation coefficient with time while Aqua shows a positive one. Terra AOD decreases

0.0071/year, while Aqua increases 0.0015/year. Aqua have lower correlation coefficient for AOD compared to Terra, which indicates Aqua performed more stable during the study period. p.11, lines 298-300 new paragraph added In order to understand whether the discrepancy temporal trend of Terra and Aqua is a result of regional conditions or if it exists in all sites, we investigated Terra, Aqua, MISR, and AERONET over other sites. p.11, lines 302-303 paragraph has been modified Both MODIS/Aqua and MODIS/Terra AOD show a stable trend over time at Mezaria site (not shown in the figure) with a correlation coefficient of 0.11 and 0.04 respectively. Both Terra and Aqua AOD increase 0.008 and 0.001/year, respectively. p.12, lines 310-314, new paragraph added where Terra AOD decreases 0.0027/year, while Aqua increases 0.0066/year. Although Solar Village, Mezaria, and Bahrain are all located in or next to a desert region, the inconsistency between Terra and Aqua measurements is subject to the regional conditions. For example, the large water body surrounding Bahrain could mean that the great majority of the MODIS retrievals are from Dark Target algorithm. p.12, lines 316-327, new paragraph added Over Cairo, MODIS/Terra, MODIS Aqua, and MISR measurements agree on AOD increase by 0.001, 0.0007, and 0.0007/year respectively with correlation coefficients 0.10, 0.04, and 0.22 respectively. Despite the deviation between the three aforementioned sensors, they all agree on AOD temporal trend increase over Cairo. This could be attributed to the high pollution level at the mega city of Cairo due to high population, vehicle emission, and biomass burning. Taman site (Fig. 4c): MISR AOD agrees with Taman AERONET on a positive trend indicating the efficiency of MISR V22 algorithm over green areas with less black carbon particles. Aqua measurements show temporal AOD decrease of 0.0079/year with a correlation coefficient of 0.81 and Terra show AOD decrease of 0.0043/year with a correlation coefficient of 0.35. Meanwhile, MISR shows AOD increase of 0.0014/year with a correlation coefficient of 0.19.

Comments from Referees I noticed the short comment by Andrew Sayer (I usually try to avoid reading other reviews in discussion journals, but as a comment on data versions this seemed to be a particularly relevant point), and I agree that it is vital that the most up-to-date data versions are used for all three of the datasets. If the current versions are not used then the analysis in this paper is of only minimal historical interest. Therefore please make sure that you are using the new Version 3 AERONET products, for example. I do not know how much difference to the results re-performing the analysis will cause, but presumably there will be differences in almost all of the figures and tables. Author's response We would like to confirm that we have used Level 2.0 Version 3 AERONET data available at https://aeronet.gsfc.nasa.gov. This has been highlighted in the paper at p.6, lines 48-49. For MODIS data, we have used Collection 6.1. Both dark target and deep blue algorithms have been used. Dark target retrievals were used over water regions while deep blue data were used over land. Data are available at https://giovanni.gsfc.nasa.gov/giovanni. For MISR data, we have choose to use V22 rather than V23, released on February 12, 2018, in our analysis because of few know issues know with this product that are still under formal validation. Some of these known issues are directly related to data reliability over bright surfaces compared to dark water, which is significant for our study. We have responded to Andrew Sayer through public discussion to explain that for the results reliability we should not use V23 MISR data for this study. Only after these known issues are resolved, it will be more feasible to relay on the new data product. Below please find our detailed response to Andrew Sayer Dear Andrew, Thank you very much for the short comment regarding the data version used in the article.

MISR Indeed, we are aware of version 23 (V23) MISR data released on February 12, 2018, however few known issues with the new product are still under formal validation. Some of these known issues are related to data reliability over bright surfaces compared to dark water, which is significant for our study. Moreover, we have found that changes in the new product has no significant impact on the results presented in our article as explained below in major and minor differences between V23 and V22 MISR product. To ensure data reliability based on known issues and insignificant impact of the new product on our results, we preferred to use the most recent V22 in our analysis.

Major differences between V23 and V22 MISR products 1- Initial assessments of the results from the 4.4 km resolution V23 retrieval algorithm show that V22 AOD retrievals perform similar to V23 relative to AERONET. V23, however perform significantly better than V22 only relative to high spatial density AERONET Distributed Regional Aerosol Gridded Observation Network (DRAGON) deployments which is out of the scope of our study. 2- V22 has similar performance as V23 in reporting non-spherical aerosols in places where they are climatologically expected, particularly when the AOD is large. Both versions effectively discriminates small, medium, and large particles in exactly similar pattern. 3- Although V23 added AOD grid points below 0.025, which eliminates gap at low AODs, observed relative to AERONET, this update should not affect the results in our article, as we are not dealing with such low AOD values. 4- V23 changes in the snow-ice mask source by applying a more conservative cloud screening logic. This should have no effect on the results presented in our paper as we have performed our comparative analysis mostly over an arid/semi-arid region. 5- V23 change in near-surface wind speed source has no significant effect on our results as only the total wind speed is used in the dark water aerosol retrievals; this change does not affect the Aerosol Product. 6- V23 added a correction factor to take into consideration the effect of chlorophyll ("underlight") on MISR red and NIR bands over Dark Water. This reduces AODs retrieved over dark water; however, its significantly affect low AODs values only. Minor differences between V23 and V22 MISR products 1- Significant field name and content changes in V23 relative to V22, which makes the product significantly more accessible. This however has no effect on the results discussed in our article. 2- Switch from HDF4, stacked-block format to NetCDF-4 conventional format. This however has no effect on the results discussed in our article. 3- Provide per-retrieval geolocation and time information to make product easier to use. This also has no effect on the results presented. If you still believe that the new data product could significantly change the results taking into consideration possible AOD range at the study region, please let me know and we can definitely check the results against the new version. AERONET For the AERONET data, we have used Level 2.0 Version 3 available at https://aeronet.gsfc.nasa.gov. We will highlight this in the article. MODIS For MODIS

data, we have used Collection 6.1. Both dark target and deep blue algorithms have been used. Dark target retrievals were used over water regions while deep blue data were used over land. Data are available at https://giovanni.gsfc.nasa.gov/giovanni. We will highlight this in the article. p.4 Lines 110 – 114 have been added. Comments from Referees Please also clarify whether you are using the Dark Target (DT) and/or the Deep Blue (DB) AOD retrievals, since these use very different retrieval methods and it is a vital distinction to make. Presumably, the MODIS AODs over central desert sites such as Solar Village or Tamanrasset would be from the Deep Blue algorithm, while coastal sites such as Bahrain would have a greater prevalence of DT retrievals. It would perhaps make more sense to discriminate the MODIS AODs further, between retrievals using the DT and the DB algorithms. A possible question might be whether the DB or the DT algorithm performs better in the vicinity of Bahrain or other such sites on the desert margins? Author's response Both dark target and deep blue algorithms have been used. Dark target retrievals were used over water regions while deep blue data were used over land. Data are available at https://giovanni.gsfc.nasa.gov/giovanni. For regions like Bahrain where large water body surrounds land, a combined Dark Target and Deep Blue AOD product for land and Ocean has been used. The product is available through https://giovanni.gsfc.nasa.gov/giovanni. p.6, Lines 145- 149 was added.

Specific Comments Comments from Referees p.2, lines 36-37: why is this in italics? Author's response Italics format has been removed. Author's changes in manuscript p.2, lines 36-37: Italics format has been removed.

Comments from Referees Section 2.2: if MODIS Deep Blue retrievals are used (and they should be), please also describe them here Author's response The author would like to confirm that both dark target and deep blue algorithms have been used. Dark target retrievals were used over water regions while deep blue data were used over land. Data are available at https://giovanni.gsfc.nasa.gov/giovanni. The Deep Blue retrievals have been described on section 2.2 P5 L 127 - 132 The Deep Blue is a NASA developed algorithm to calculate AOD over land using MODIS data. Bu measuring contrast between aerosols and surface features, Deep Blue retrieves AOD. Over bright land, Deep Blue uses (0.412, 0.470/0.490 $\mu$m) and dark land (0.470/0.490, 0.650 $\mu$m) for AOD retrievals. Over water, the Deep Blue algorithm is not used. The MODIS dark-target algorithm is designed aerosol retrieval from MODIS observations, over ocean (dark in visible and longer wavelengths) and dark land surfaces (low values of surface reflectance) (e.g., dark soil and vegetated regions) in parts of the visible (VIS, 0.47 and 0.65 $\mu$m) and shortwave infrared (SWIR, 2.1 $\mu$m) spectrum (Kaufman et al., 1997). Author's changes in manuscript New paragraph has been added to section 2.2 to describe Deep Blue algorithm P5 L 127 - 132

Comments from Referees Throughout the manuscript there are language issues which should be corrected Author's changes in manuscript Thank you for the comment. We have carefully reviewed the English through the manuscript and the following corrections have been made: Line 11: comma inserted after MISR Line 15: grammar correction: MODIS/terra AOD indicates instead of indicate Line 33: sentence revised to: "that has major effects on human activities in the Arabian" Lines 42-43: revised for clarity. Line 121: sentence rephrased for clarity Line 136-137: sentence rephrased for clarity Line 147-149: sentence rephrased for clarity Table 2 caption has been modified p.28 Lines 838-840

Comments from Referees p.14, line 330: do you know what these peaks indicate? On brief speculation I might imagine that the first peak is indicative of industrial aerosol and the second peak might be indicative of dust. Ångström coefficient values may give some evidence as to what these might be. Author's response P14, lines 332-335 have been added Ångström exponent (AE), dependency of the AOD on wavelength, can also be used to determine particles' size where the smaller the particle the larger the exponent. AE analysis show that the first peak at 0.25 is indicative of industrial particles with high AE values and the second peak at 0.35 indicates dust aerosol. High anthropogenic loading could be attributed to rapidly growing aluminum industry in Bahrain (Farahat 2016).
Comments from Referees p.15, lines 397-400: if the MODIS retrievals are preferentially coming from the Gulf, does that mean that the great majority of the retrievals over Bahrain are from DT?

Author's response The MODIS matched AERONET data are averaged from measurements that are within a radius of about 27.5 km from the AERONET station and within 30 min of the satellite flyover the station. For such a small country like Bahrain surrounded with a large water area, MODIS retrievals are preferentially coming from the water. Combined Dark Target and Deep Blue products are used for Bahrain the majority of the measurement are from DT.

Comments from Referees p.14, line 253: 'topology'. I think you mean 'topography'?

Author's response Thank you. 'topology' has been replaced with 'topography'

Author's changes in manuscript p.16, line 419

Please also note the supplement to this comment:
https://www.ann-geophys-discuss.net/angeo-2018-79/angeo-2018-79-AC2-supplement.pdf

**Supplement:**

[revised manuscript text omitted]

version 23 (V23) retrievals, released on February 2018, was not used in this study, as it has few known issues with the new product that are still under formal validation. Some of these known issues are related to data reliability over bright surfaces compared to dark water, which is significant for our analysis (Garay et al., 2018).

**2.2 MODIS**

The Moderate Resolution Imaging Spectroradiometer (MODIS) is a payload instrument on board the Terra and Aqua satellites. Terra's and Aqua orbit around the Earth from North to

South and South to North across the equator during the morning and afternoon respectively (Kaufman et al., 1997). Terra MODIS and Aqua MODIS provides nearly daily coverage of the Earth's surface and atmosphere in 36 wavelength bands, ranging from 0.412 to 41.2 µm, with spatial resolutions of 250 m (bands 1-2), 500 m (bands 3-7), 1000 m (bands 8-36).

Located near-polar orbit (705 km), MODIS has swath dimensions of 2330 km × 10 km and a scan rate of 20.3 rpm.  With its high radiometric sensitivity and swath resolution MODIS

retrievals provides information about aerosols optical and physical characteristics. MODIS

uses 14 spectral band radiance values to evaluate atmospheric contamination and determine whether scenes are affected by cloud shadow (Ackerman et al., 1998).

The Deep Blue is a NASA developed algorithm to calculate AOD over land using MODIS

data. Bu measuring contrast between aerosols and surface features, Deep Blue retrieves

AOD.  Over bright land, Deep Blue uses (0.412, 0.470/0.490 µm) and dark land (0.470/0.490,

0.650 µm) for AOD retrievals. Over water, the Deep Blue algorithm is not used.

The MODIS dark-target algorithm derives aerosol characteristics, including AOD, over ocean (dark in visible and longer wavelengths) and dark land surfaces (low values of surface reflectance) (e.g., dark soil and vegetated regions) in parts of the visible (VIS, 0.47 and 0.65

µm) and shortwave infrared (SWIR, 2.1 µm) spectrum (Kaufman et al., 1997).

| | |
|---|---|
| **Deleted:** | is designed aerosol retrieval from MODIS observations |
| **Deleted:** | , |
| **Deleted:** | |

Level 2 collection 6.1 of the algorithm are used to retrieve MODIS aerosols' time series data. Levy *et al.* (2010) reported that the dark-target algorithm AOD at 550 nm measurement for (C005) includes uncertainty of $\pm (0.05\tau+0.03)$ and $\pm (0.15\tau+0.05)$ over ocean and land respectively. This uncertainty is caused by uncertainties in computing cloud masking, surface reflectance, aerosol model type (e.g., single scattering albedo), pixels selections and instrument calibration. Both dark target and deep blue algorithms have been used. Dark target retrievals were used over water regions while deep blue data were used over land. Data are available at https://giovanni.gsfc.nasa.gov/giovanni. For regions like

Bahrain where large water body surrounds land, a combined Dark Target and Deep Blue

AOD for land and Ocean has been applied.

**2.3 AERONET**

The Aerosol Robotic Network (AERONET) Holben et al., 1998 and Holben et al., 2001 is a ground-based remote sensing aerosols network that provides a long-term data related to aerosol optical, microphysical and radiative properties. With over 700 global stations, the

AERONET data is widely used in validating satellite retrievals Chu et al., 1998 and Higurashi et al., 2000.

The sun photometers used by AERONET include sun collimators to measure spectral directbeam solar radiation. The collimators are used to determine columnar spectral AOD and water vapour, provided at a temporal resolution of approximately 10–15 min (Sayer et al.

2014). AERONET direct-sun AOD has a typical uncertainty of 0.01–0.02 (Holben et al.,

1998) and is provided at multiple wavelengths at 340, 380, 440, 500, 675, 950, and 1020 nm.

Seven AERONET sites were selected for MODIS/ Terra, MODIS/ Aqua, and MISR/Terra satellites validation in this study (Table 1.). The sites were selected based on their geographic locations to represent aerosols characteristics over North Africa and the Middle East (Farahat et al., 2016). A record of long-term data collection was another factor in the selection process.

Level 2.0 Version 3 AERONET data available at https://aeronet.gsfc.nasa.gov have been used in the study.

**Data Matching Approach**

Multi-sensors data matching approach requires using only spatial and temporal matching data to reduce uncertainties associated with using different instruments and clouds shadow 
[revised manuscript text omitted]

Trends of aerosol loading from 2000 to 2005 are analysed by plotting fitting lines of monthly mean AOD retrievals by MISR and MODIS/Terra and Aqua. The AOD retrieved by different instrument shows different trends. MODIS/ Aqua and MISR AOD at Solar Village have positive trends, while MODIS/ Terra AOD have negative trends along time series (Fig. 4a). Terra depicts a negative correlation coefficient with time while Aqua shows a positive one. Terra AOD decreases 0.0071/year, while Aqua increases 0.0015/year. Aqua have lower correlation coefficient for AOD compared to Terra, which indicates Aqua performed more stable during the study period. Discrepancy between Aqua and Terra retrievals could be related to instrument calibration, or the difference in aerosol and cloud conditions from the morning to the afternoon. Both MODIS Aqua and Terra are underestimating AOD at Solar Village. MISR AOD trend shows a better agreement with Solar Village AERONET AOD as compared to MODIS.

In order to understand whether the discrepancy temporal trend of Terra and Aqua is a result of regional conditions or if it exists in all sites, we investigated Terra, Aqua, MISR, and AERONET over other sites.

Both MODIS/Aqua and MODIS/Terra AOD show a stable trend over time at Mezaria site (not shown in the figure) with a correlation coefficient of 0.11 and 0.04 respectively. Both Terra and Aqua AOD increase 0.008 and 0.001/year, respectively. Aqua AOD over Bahrain (not shown in the figure) show, less time trend stability compared to those at Solar Village with a correlation where Terra AOD decreases 0.0027/year, while Aqua increases

0.0066/year. Although Solar Village, Mezaria, and Bahrain are all located in or next to a desert region, the inconsistency between Terra and Aqua measurements is subject to the regional conditions. For example, the large water body surrounding Bahrain could mean that the great majority of the MODIS retrievals are from Dark Target algorithm. MODIS/Aqua,

MODIS/Terra, and MISR AODs depicts a positive trend over Cairo, however a 2 years of available AERONET data is not sufficient for the trend analysis (Fig. 4b). Over Cairo,

MODIS/Terra, MODIS Aqua, and MISR measurements agree on AOD increase by 0.001,

0.0007, and 0.0007/year respectively with correlation coefficients 0.10, 0.04, and 0.22

respectively. Despite the deviation between the three aforementioned sensors, they all agree on AOD temporal trend increase over Cairo. This could be attributed to the high pollution level at the mega city of Cairo due to high population, vehicle emission, and biomass burning.

Taman site (Fig. 4c): MISR AOD agrees with Taman AERONET on a positive trend indicating the efficiency of MISR V22 algorithm over green areas with less black carbon particles. Aqua measurements show temporal AOD decrease of 0.0079/year with a correlation coefficient of 0.81 and Terra show AOD decrease of 0.0043/year with a correlation coefficient of 0.35. Meanwhile, MISR shows AOD increase of 0.0014/year with a correlation coefficient of 0.19.

Long-range (2000 – 2015) tendency indicates that contradictory AOD trend of Terra and

Aqua is individually explicit for each site and does not necessarily apply everywhere.

AOD difference between Terra and Aqua could be used as another indicator of the long- range satellites performance. AOD difference (Terra AOD minus Aqua AOD) varies from -

0.01 to 0.19, -0.10 to 0.18, -0.02 to 0.13 over Solar Village, Taman, and Cairo respectively (Fig. 5). Over the Solar Village, Terra overestimates AODs during 2002-2004 and underestimates the AOD after 2005. Although Cairo and Taman show similar trend however over/underestimation amount is not unique for all sites. This is an indication that Aqua and

Terra retrievals disagreement takes place regardless of the region but site sampling has significant effect on the amount of contradiction.

Statistical comparison between MISR and MODIS/Terra AODs at corresponding

AERONET stations is performed by calculating Gfraction using of $\Delta\tau = \pm0.05 \pm 0.15\tau_{AERO}$

as a confidence interval. Over the region 1, MISR AODs retrievals are more accurate than

[revised manuscript text omitted]

in the matched data set but misses the peak at 0.5. Ångström exponent (AE), dependency of the AOD on wavelength, can also be used to determine particles' size where the smaller the particle the larger the exponent. AE analysis show that the first peak at 0.25 is indicative of industrial particles with high AE values and the second peak at 0.35 indicates dust aerosol.

The MISR climatology agrees well with the AERONET all data climatology for all AODs.

MODIS on the other hand shows an extremely large frequency of AODs at 0.1 not represented by AERONET coupled with an underestimation of AODs greater than 0.3. This could be attributed to the size of the matching window and MODIS retrievals preferentially coming from the Arabian Gulf.

SAADA station is located close to some hiking trails at the Agoundis Valley in the Atlas

Mountains about 197 km from the city of Marrakesh.

MISR AODs matched to AERONET agree well with MISR full climatology retrievals over

SAADA station. Both retrievals slightly underestimate SAADA full climatology and over estimates SAADA matched data retrievals at AODs equal to 0.1 while show good agreement for AODs greater than 0.1.  MODIS matched to AERONET retrievals overestimate the frequency of AODs greater than 0.3. While MODIS AODs matched to AERONET captures climatology at AODs between 0.2 to 0.25, AODs frequency retrievals are under-sampled at

AODs between 0.1 to 0.15 with about 13 % less events than SAADA all data retrievals at

AODs equal to 0.1.

Figure (9a, b) indicates right skewed distribution of SAADA AODs towards small AOD

values with 11.5 % and 30.1 % of AODs > 0.4 as measured by MISR and MODIS

respectively. Taking into consideration MODIS overestimation we conclude that SAADA

site is characterized by small AODs values and this could be related to the land topography where the station is located.

While MISR is capturing high AODs climatology over SAADA, both MISR and MODIS

are underestimating the frequency of lower AODs events. Nevertheless, MISR captures the climatology of AODs less than 0.1 missed by MODIS retrievals.

Taman AERONET station is located at the oasis city of Tamanrasset, which lies in Ahaggar

National Park at southern Algeria.

Figure (10 a, b) depicts that Taman AERONET AOD climatology is similar to those at

SAADA and has a high frequency of low AODs events. Both MISR AODs matched to

AERONET and MISR all data do not well capture the frequency of AODs less than 0.1 or larger than 1 while well describe the climatology for AODs in the range of 0.1 to 1. MODIS

AODs matched data to AERONET correctly describe climatology with slight overestimation of AODs frequencies between 0.05 – 0.15 while not capturing AODs frequencies greater than 1. MISR and MODIS show similar prominent peaks at 0.1, 0.25, and 0.35, not observed in Taman AERONET AOD climatology, with more peaks observed by MISR at 0.5, 0.6, and

0.8. Average AODs in SAADA and Taman is ~ 50 percent less than observed at Solar

Village, Mezaria, and Bahrain sites.

Except for AODs greater than 1 where ground observations could be more robust, both MISR

and MODIS retrievals can provide very good climatology matching over Taman site.

Taking into consideration lower number of MISR matching AERONET observations compared to MODIS ~ 33 and 43 percent over SAADA and Taman respectively, MISR is outperforming over these two sites, which can be attributed to its multiangle viewing capabilities over complex terrains including mountainous areas (Atlas Mountains).

Cairo is a mega city well known for its high pollution due to traffic and agriculture activities.

MISR and MODIS matched data correctly capture AOD climatology over Cairo compared to AERONET as shown in Figure (11a, b). MISR retrievals collocated with AERONET

capture prominent peaks of AERONET AOD at 0.15 – 0.25 and 0.5 with small underestimation observed at 0.3. MISR 'all data' AOD climatology over Cairo station agrees better with AERONET AOD climatology vs. collocated dataset with some oversampling at

0.15. Frequency of high AODs retrievals at 0.7 and 0.8 have not been captured by MISR

matched or all data retrievals. MODIS matched to AERONET AODs are also able to well present Cairo climatology data with a high overestimation of AODs frequency between 0.05

- 0.2 and an underestimation of AODs larger than 0.4.

[revised manuscript text omitted]

**Tables' caption**

Table 1. Geographic location of the AERONET sites used in this study

Table 2. Statistics for the calculation of MODIS/Terra, MODIS/Aqua, and MISR with that of AERONET measurements over seven sites in the Middle East and North Africa, including R: correlation coefficient, Gfraction: good fraction; N: number of observations

Table 3. Statistics for biomass and mixed sites, parameters as in Table 3. Caption.

Table 4. Percentage of AODs retrievals greater than 0.4 recorded by AERONET all data,

MISR all data and MODIS matched data over seven AERONET sites in Middle East and

North Africa.

**Figures caption**

Figure 1. Location of the AERONET stations over North Africa and the Middle East. The numbers on the map indicate the site location as 1: Saada, 2: Tamanrasset_INM, 3: Cairo,

4: Sede Boker, 5: Solar Village, 6: Mezaira, 7: Bahrain.

Figure 2. Scatter plot of MISR AOD versus AERONET AOD based on seasons and aerosols categorization.

Figure 3. Scatter plot of MODIS AOD versus AERONET AOD based on seasons and aerosols categorization.

Figure 4. Time series of monthly mean AOD derived from MODIS/Aqua, MODIS/Terra,

MISR and AERONET over a) dust b) biomass and c) mixed dominated aerosol regions.

Figure 5. Long range AOD difference for MODIS/Terra and MODIS/Aqua over the dust, biomass and mixed sites.

Figure 6. Histogram of the MISR, MODIS and Solar Village AERONET measurements a)

MISR b) MODIS data retrievals.

Figure 7. Histogram of the MISR, MODIS and Mezaria AERONET measurements a)

MISR b) MODIS data retrievals.

Figure 8. Histogram of the MISR, MODIS and Bahrain AERONET measurements a) MISR

b) MODIS data retrievals.

Figure 9. Histogram of the MISR, MODIS and SAADA AERONET measurements a)

MISR b) MODIS data retrievals.

Figure 10. Histogram of the MISR, MODIS and Taman AERONET measurements a)

MISR b) MODIS data retrievals.

Figure 11. Histogram of the MISR, MODIS and SEDEE Boker AERONET measurements a) MISR b) MODIS data retrievals.

Figure 12. Histogram of the MISR, MODIS and Cairo AERONET measurements a) MISR

b) MODIS data retrievals.

| Location name | Lon./Lat. | Measurement period |
|---|---|---|
| Solar Village | 24.907º N/46.397º E | 2000-2015 |
| Mezaria | 23.105º N/53.755º E | 2004-2015 |
| Bahrain | 26.208º N/50.609º E | 2000-2006 |
| Saada | 31.626º N/8.156º W | 2003-2015 |
| Taman | 22.790º N/5.530º E | 2000-2015 |
| Cairo | 30.081º N/31.290º E | 2005 -2007 |
| Sede Boker | 30.855º N/34.782 º E | 2000-2015 |

Table 1.

Table 2.

| AERONET Site | Sensor | Season | Mean Value | | N | R | Gfraction (%) |
|---|---|---|---|---|---|---|---|
| | | | AERONET | Satellite | | | |
| | MISR | DJF | 0.31±0.22 | 0.38±0.20 | 338 | 0.94 | 60.05 |
| | | MAM | 0.39±0.27 | 0.45±0.23 | 89 | 0.94 | 65.16 |
| | | JJA | 0.39±0.18 | 0.45±0.17 | 141 | 0.90 | 70.21 |
| | | SON | 0.27±0.16 | 0.35±0.14 | 3 | 0.99 | 33.33 |
| Solar Village | | DJF | 0.27±0.19 | 0.33±0.17 | 1500 | 0.48 | 51.80 |

| AERONET Site | | Season | Mean Value | | N | R | Gfraction (%) |
|---|---|---|---|---|---|---|---|
| | | | | | | | |
| | **MODIS Terra** | **MAM** | 0.36±0.24 | 0.26±0.17 | 389 | 0.68 | 90.23 |
| | | **JJA** | 0.34±0.17 | 0.42±0.19 | 429 | 0.41 | 54.31 |
| | | **SON** | 0.22±0.10 | 0.36±0.12 | 471 | 0.51 | 28.87 |
| **Mezaria** | **MISR** | **DJF** | 0.33±0.15 | 0.40±0.17 | 60 | 0.89 | 75.00 |
| | | **MAM** | 0.32±0.19 | 0.41±0.22 | 13 | 0.90 | 69.23 |
| | | **JJA** | 0.42±0.13 | 0.47±0.17 | 21 | 0.85 | 80.95 |
| | | **SON** | 0.29±0.07 | 0.36±0.07 | 22 | 0.87 | 77.27 |
| | **MODIS Terra** | **DJF** | 0.32±0.15 | 0.35±0.19 | 198 | 0.86 | 74.74 |
| | | **MAM** | 0.44±0.33 | 0.45±0.27 | 115 | 0.92 | 78.07 |
| | | **JJA** | 0.39±0.14 | 0.43±0.20 | 89 | 0.81 | 71.91 |
| | | **SON** | 0.28±0.13 | 0.30±0.16 | 97 | 0.87 | 77.31 |
| **Bahrain** | **MISR** | **DJF** | 0.37±0.11 | 0.31±0.10 | 17 | 0.73 | 100 |
| | | **MAM** | 0.31±0.11 | 0.28±0.14 | 3 | 0.89 | 100 |
| | | **JJA** | 0.40±0.09 | 0.36±0.09 | 8 | 0.69 | 100 |
| | | **SON** | 0.40±0.09 | 0.30±0.05 | 4 | 0.98 | 100 |
| | **MODIS Terra** | DJF | 0.42±0.29 | 0.20±0.19 | 121 | 0.41 | 93.38 |
| | | MAM | 0.50±0.28 | 0.13±0.15 | 25 | 0.26 | 96.00 |
| | | JJA | 0.55±0.26 | 0.31±0.27 | 42 | 0.50 | 88.09 |
| | | SON | 0.35±0.14 | 0.21±0.12 | 29 | 0.32 | 93.10 |

                                    Table 3.

| AERONET Site | Sensor | Season | Mean Value | | N | R | Gfraction (%) |
|---|---|---|---|---|---|---|---|
| | | | AERONET | Satellite | | | |
| | **MISR** | **DJF** | 0.24±0.16 | 0.22±0.15 | 149 | 0.93 | 97.29 |
| | | **MAM** | 0.21±0.13 | 0.19±0.11 | 53 | 0.89 | 96.15 |
| | | **JJA** | 0.29±0.14 | 0.27±0.15 | 80 | 0.93 | 97.46 |
| | | **SON** | 0.19±0.15 | 0.19±0.12 | 60 | 0.94 | 98.30 |

| | | | | | | | |
|---|---|---|---|---|---|---|---|
| **SAADA** | | **DJF** | 0.23±0.16 | 0.32±0.21 | 550 | 0.57 | 57.81 |
| | | **MAM** | 0.24±0.18 | 0.39±0.23 | 90 | 0.43 | 44.44 |
| | **MODIS** | **JJA** | 0.30±0.17 | 0.45±0.18 | 201 | 0.40 | 45.27 |
| | **Terra** | **SON** | 0.19±0.13 | 0.22±0.14 | 162 | 0.71 | 72.39 |
| **Taman** | | **DJF** | 0.19±0.23 | 0.24±0.19 | 135 | 0.92 | 70.89 |
| | | **MAM** | 0.29±0.22 | 0.35±0.24 | 24 | 0.97 | 82.60 |
| | **MISR** | **JJA** | 0.35±0.30 | 0.39±0.19 | 36 | 0.85 | 71.42 |
| | | **SON** | 0.19±0.15 | 0.19±0.12 | 60 | 0.94 | 98.30 |
| | | **DJF** | 0.19±0.22 | 0.18±0.16 | 319 | 0.67 | 81.81 |
| | **MODIS** | **MAM** | 0.24±0.19 | 0.22±0.17 | 67 | 0.55 | 83.58 |
| | **Terra** | **JJA** | 0.37±0.32 | 0.29±0.20 | 69 | 0.69 | 84.05 |
| | | **SON** | 0.14±0.14 | 0.13±0.10 | 117 | 0.54 | 84.61 |
| **Cairo** | | **DJF** | 0.33±0.20 | 0.28±0.11 | 13 | 0.94 | 100 |
| | | **MAM** | 0.22±0.06 | 0.24±0.08 | 5 | 0.99 | 100 |
| | **MISR** | **JJA** | 0.43±0.23 | 0.34±0.11 | 5 | 0.99 | 100 |
| | | **SON** | 0.38±0.21 | 0.29±0.12 | 4 | 0.97 | 100 |
| | | **DJF** | 0.33±0.16 | 0.20±0.11 | 158 | 0.30 | 95.56 |
| | **MODIS** | **MAM** | 0.32±0.16 | 0.12±0.08 | 39 | 0.25 | 100 |
| | **Terra** | **JJA** | 0.35±0.14 | 0.28±0.07 | 58 | 0.17 | 94.82 |
| | | **SON** | 0.38±0.19 | 0.20±0.09 | 29 | 0.07 | 93.82 |
| **SEDEE_BOKER** | | **DJF** | 0.14±0.06 | 0.21±0.07 | 23 | 0.87 | 40.90 |
| | | **MAM** | 0.14±0.05 | 0.24±0.09 | 13 | 0.68 | 33.33 |
| | **MISR** | **JJA** | 0.16±0.05 | 0.24±0.06 | 163 | 0.85 | 33.33 |
| | | **SON** | 0.15±0.07 | 0.23±0.06 | 72 | 0.89 | 33.80 |
| | | **DJF** | 0.16±0.12 | 0.23±0.14 | 1312 | 0.36 | 53.50 |
| | **MODIS** | **MAM** | 0.21±0.18 | 0.24±0.19 | 338 | 0.34 | 65.68 |
| | **Terra** | **JJA** | 0.16±0.09 | 0.33±0.13 | 392 | 0.27 | 17.34 |
| | | **SON** | 0.16±0.09 | 0.23±0.12 | 477 | 0.46 | 58.49 |

Table 4.

¶
¶
¶
¶
¶
¶
¶
¶
¶
¶
¶

| | AERONET | | MISR | | MODIS | |
|---|---|---|---|---|---|---|
| | | AOD | | AOD | | AOD |
| | N | % > 0.4 | N | % > 0.4 | N | % > 0.4 |
| Solar Village | 3978 | 28.7 | 684 | 32.8 | 2789 | 30.1 |
| Mezaria | 1650 | 30.2 | 547 | 45.7 | 498 | 40.7 |
| Bahrain | 1117 | 33.3 | 676 | 35.7 | 217 | 18.4 |
| SAADA | 3184 | 10.8 | 667 | 11.5 | 1004 | 34.6 |
| Taman | 1863 | 17.9 | 845 | 22.6 | 572 | 9.4 |
| Cairo | 269 | 53.5 | 620 | 17.7 | 284 | 4.2 |
| SEDEE | 5722 | 4.8 | 675 | 9 | 2519 | 12.8 |

[Figure]

Figure 1.

[Figure]

**Region 1**          **Region 2**

[Figure]

¶
¶
¶

[Figure]

                              Figure 2.

          **Region 1**                    **Region 2**

[Figure]

Figure 3.

[Figure]

Figure 4.

[Figure]

Figure 5.

[Figure]

[Figure]

Figure 6.

[Figure]

[Figure]

Figure 7.

[Figure]

[Figure]

Figure 8.

¶
¶

[Figure]

[Figure]

Figure 9.

[Figure]

[Figure]

Figure 10.

[Figure]

[Figure]

Figure 11.

[Figure]

[Figure]

Figure 12.

---

## Author Comment (AC3) · 19 Sep 2018

Response to Anonymous Referee # 2 Dear Referee, Thank you very much for your feedback about our article. We greatly appreciate the comments. We have addressed all your comments and we have revised the manuscript accordingly. For your consideration, we have included a copy of the revised article with track changes. Please find below our response to your comments. Regards, Ashraf Farahat

Comments from Referees The author of this manuscript has done quite interesting work, well analyzed "Comparative Analysis of MODIS, MISR 1 and AERONET Climatology over the Middle East and North Africa". In general the manuscript is interesting and well written. The results have been presented and discussed well and thoroughly.

[Figure]

In my opinion, the topic discussed in this paper is suitable for publication. Overall I recommend acceptance of this paper for publication with minor revisions. Please see the specific comments below. Author's response We would like to thank the reviewer very much for his/her comments and for recommending the publication of our article with minor revision. We have addressed all the reviewer comments below. Comments from Referees Line 11: please insert comma after MISR Author's response Done Comments from Referees Line 15: please check the grammar, i.e. MODIS/terra AOD indicates instead of indicate Author's response Done Comments from Referees Line 33: please use like this "that has major effects on human activities in the Arabian" Author's response Done Comments from Referees Line 42-43: please make it clear to the reader Author's response p.2 Lines 42-43 have been modified Aerosol optical depth, AOD, is a parameter to measure the extinction of a beam of light as it passes through a layer of atmosphere that contains aerosols.

Comments from Referees Line 121: please rephrase the sentence. Author's response p. 5 Lines 131-132 (previous 121 – 124) have been rephrased. The MODIS dark-target algorithm derives aerosol characteristics, including AOD, over ocean (dark in visible and longer wavelengths) and dark land surfaces (low values of surface reflectance) (e.g., dark soil and vegetated regions) in parts of the visible (VIS, 0.47 and 0.65 $\mu$m) and shortwave infrared (SWIR, 2.1 $\mu$m) spectrum (Kaufman et al., 1997).

Comments from Referees Line 136-137: please rephrase the sentence Author's response p. 6 Lines 156-159 (previous 136 -138) have been rephrased The sun photometers used by AERONET include sun collimators to measure spectral direct-beam solar radiation. The collimators are used to determine columnar spectral AOD and water vapour, provided at a temporal resolution of approximately 10–15 min (Sayer et al. 2014).

Comments from Referees Line 142: please mention the name of satellite Author's response The names of the satellites are now mentioned p.6 L157-158 (previous L 142) Seven AERONET sites were selected for MODIS/ Terra, MODIS/ Aqua, and

MISR/Terra satellites validation in this study (Table 1.).

Comments from Referees Line 147-149: please revise the sentence. Author's response p.7 Lines 174 – 176 have been revised (previous 147-149). Multi-sensors data matching approach requires using only spatial and temporal matching data to reduce uncertainties associated with using different instruments and clouds shadow Liu and Mishchenko (2008) and Mishchenko et al., 2009.

Comments from Referees Line 158: The authors have mentioned that they have used second approach in this study. Why did the authors not use the first approach? Author's response Both approaches have their limitations; however, we used (Mishchenko et al., 2010 approach) as it simultaneously matches location and time between the AERONET station and satellites. This certainly reduces the number of available matched data points; however, it eliminates data uncertainty compared to the other approach.

Comments from Referees Line 176: The authors have used only two statistics parameters to validate the satellite data. It is suggested to use more parameters for the validation. It is also observed that authors have not mention the value of statistical parameters in the figures.

Author's response We totally agree with the referee comments that more statistical parameters would strength the validation process. Indeed, we have tried to use fours statistical parameters namely relative error, correlation coefficient, root mean square deviation, and good fraction. That said, for our specific study we found that the same conclusion can be approached using only two parameters. In order to avoid lengthy tables and redundancy that may confuse readers, we decided to present two parameters only in the tables. We have presented some of the statistical parameters in the figures, the rest are listed in Tables 1-4.

Comments from Referees Line 196: please correct number of equations in the text. Author's response p.8 Lines 223, and 224. (Previous Line 196). Thank you. Equation numbers are now corrected.

Comments from Referees Table 2: Caption of table should be precise and general and table value should match according to the caption e.g RMSE is mentioned in the caption but not presented in the table, G-fraction and Gfraction should same in the text. Author's response Table 2 caption has been modified p.28 Lines 843-845 Table 2. Statistics for the calculation of MODIS/Terra, MODIS/Aqua, and MISR with that of AERONET measurements over seven sites in the Middle East and North Africa, including R: correlation coefficient, Gfraction: good fraction; N: number of observations

We have also used "Gfraction" all over the text.

Comments from Referees Table 3: Like statistics for biomass and mixed, parameter as in table 2 (but you mentioned parameter as table 3) Author's response Thank you. Typo corrected. P.28 Lines 782

Comments from Referees Second column of each table should be same if they belongs to same category. It will confuse the reader, like in table 2, you used 'sensor' but in table 3 you changed sensor to 'method' but they are the same indeed. It will confuse the reader Author's response Thank you. "Method" has been changed to "Sensor" in Table 3 Column 2

Comments from Referees Table 4: Caption of table 4 is again confusing MISR coverage but in the body of table MODIS, MISR and AERONET are all showing their coverage Author's response Thank you. Table 4 caption has been modified to Table 4. Percentage of AODs retrievals greater than 0.4 recorded by AERONET all data, MISR all data and MODIS matched data over seven AERONET sites in Middle East and North Africa.

Comments from Referees FIGURE 1: Check the grammar of caption of figure1 e.g. "The numbers on the map indicate, not indicates" What is the source of this fig? Please combine figure 2 and 3 because they are the same actually just with different satellite data

Author's response We have corrected the grammar of figure 1 caption. We have produced the map in figure 1 in house using GIS software. We would like to thank the reviewer for his/her suggestion of combining figure 2 and figure 3 but we respectfully prefer to keep them as separate figures. Combining the two figures will make them not clear.

Please also note the supplement to this comment:
https://www.ann-geophys-discuss.net/angeo-2018-79/angeo-2018-79-AC3-supplement.pdf

**Supplement:**

[revised manuscript text omitted]

version 23 (V23) retrievals, released on February 2018, was not used in this study, as it has few known issues with the new product that are still under formal validation. Some of these known issues are related to data reliability over bright surfaces compared to dark water, which is significant for our analysis (Garay et al., 2018).

**2.2 MODIS**

The Moderate Resolution Imaging Spectroradiometer (MODIS) is a payload instrument on board the Terra and Aqua satellites. Terra's and Aqua orbit around the Earth from North to

South and South to North across the equator during the morning and afternoon respectively (Kaufman et al., 1997). Terra MODIS and Aqua MODIS provides nearly daily coverage of the Earth's surface and atmosphere in 36 wavelength bands, ranging from 0.412 to 41.2 µm, with spatial resolutions of 250 m (bands 1-2), 500 m (bands 3-7), 1000 m (bands 8-36).

Located near-polar orbit (705 km), MODIS has swath dimensions of 2330 km × 10 km and a scan rate of 20.3 rpm.  With its high radiometric sensitivity and swath resolution MODIS

retrievals provides information about aerosols optical and physical characteristics. MODIS

uses 14 spectral band radiance values to evaluate atmospheric contamination and determine whether scenes are affected by cloud shadow (Ackerman et al., 1998).

The Deep Blue is a NASA developed algorithm to calculate AOD over land using MODIS

data. Bu measuring contrast between aerosols and surface features, Deep Blue retrieves

AOD.  Over bright land, Deep Blue uses (0.412, 0.470/0.490 µm) and dark land (0.470/0.490,

0.650 µm) for AOD retrievals. Over water, the Deep Blue algorithm is not used.

The MODIS dark-target algorithm derives aerosol characteristics, including AOD, over ocean (dark in visible and longer wavelengths) and dark land surfaces (low values of surface reflectance) (e.g., dark soil and vegetated regions) in parts of the visible (VIS, 0.47 and 0.65

µm) and shortwave infrared (SWIR, 2.1 µm) spectrum (Kaufman et al., 1997).

| | |
|---|---|
| **Deleted:** | is designed aerosol retrieval from MODIS observations |
| **Deleted:** | , |
| **Deleted:** | |

Level 2 collection 6.1 of the algorithm are used to retrieve MODIS aerosols' time series data. Levy *et al.* (2010) reported that the dark-target algorithm AOD at 550 nm measurement for (C005) includes uncertainty of $\pm (0.05\tau+0.03)$ and $\pm (0.15\tau+0.05)$ over ocean and land respectively. This uncertainty is caused by uncertainties in computing cloud masking, surface reflectance, aerosol model type (e.g., single scattering albedo), pixels selections and instrument calibration. Both dark target and deep blue algorithms have been used. Dark target retrievals were used over water regions while deep blue data were used over land. Data are available at https://giovanni.gsfc.nasa.gov/giovanni. For regions like

Bahrain where large water body surrounds land, a combined Dark Target and Deep Blue

AOD for land and Ocean has been applied.

**2.3 AERONET**

The Aerosol Robotic Network (AERONET) Holben et al., 1998 and Holben et al., 2001 is a ground-based remote sensing aerosols network that provides a long-term data related to aerosol optical, microphysical and radiative properties. With over 700 global stations, the

AERONET data is widely used in validating satellite retrievals Chu et al., 1998 and Higurashi et al., 2000.

The sun photometers used by AERONET include sun collimators to measure spectral directbeam solar radiation. The collimators are used to determine columnar spectral AOD and water vapour, provided at a temporal resolution of approximately 10–15 min (Sayer et al.

2014). AERONET direct-sun AOD has a typical uncertainty of 0.01–0.02 (Holben et al.,

1998) and is provided at multiple wavelengths at 340, 380, 440, 500, 675, 950, and 1020 nm.

Seven AERONET sites were selected for MODIS/ Terra, MODIS/ Aqua, and MISR/Terra satellites validation in this study (Table 1.). The sites were selected based on their geographic locations to represent aerosols characteristics over North Africa and the Middle East (Farahat et al., 2016). A record of long-term data collection was another factor in the selection process.

Level 2.0 Version 3 AERONET data available at https://aeronet.gsfc.nasa.gov have been used in the study.

**Data Matching Approach**

Multi-sensors data matching approach requires using only spatial and temporal matching data to reduce uncertainties associated with using different instruments and clouds shadow 
[revised manuscript text omitted]

Trends of aerosol loading from 2000 to 2005 are analysed by plotting fitting lines of monthly mean AOD retrievals by MISR and MODIS/Terra and Aqua. The AOD retrieved by different instrument shows different trends. MODIS/ Aqua and MISR AOD at Solar Village have positive trends, while MODIS/ Terra AOD have negative trends along time series (Fig. 4a). Terra depicts a negative correlation coefficient with time while Aqua shows a positive one. Terra AOD decreases 0.0071/year, while Aqua increases 0.0015/year. Aqua have lower correlation coefficient for AOD compared to Terra, which indicates Aqua performed more stable during the study period. Discrepancy between Aqua and Terra retrievals could be related to instrument calibration, or the difference in aerosol and cloud conditions from the morning to the afternoon. Both MODIS Aqua and Terra are underestimating AOD at Solar Village. MISR AOD trend shows a better agreement with Solar Village AERONET AOD as compared to MODIS.

In order to understand whether the discrepancy temporal trend of Terra and Aqua is a result of regional conditions or if it exists in all sites, we investigated Terra, Aqua, MISR, and AERONET over other sites.

Both MODIS/Aqua and MODIS/Terra AOD show a stable trend over time at Mezaria site (not shown in the figure) with a correlation coefficient of 0.11 and 0.04 respectively. Both Terra and Aqua AOD increase 0.008 and 0.001/year, respectively. Aqua AOD over Bahrain (not shown in the figure) show, less time trend stability compared to those at Solar Village with a correlation where Terra AOD decreases 0.0027/year, while Aqua increases

0.0066/year. Although Solar Village, Mezaria, and Bahrain are all located in or next to a desert region, the inconsistency between Terra and Aqua measurements is subject to the regional conditions. For example, the large water body surrounding Bahrain could mean that the great majority of the MODIS retrievals are from Dark Target algorithm. MODIS/Aqua,

MODIS/Terra, and MISR AODs depicts a positive trend over Cairo, however a 2 years of available AERONET data is not sufficient for the trend analysis (Fig. 4b). Over Cairo,

MODIS/Terra, MODIS Aqua, and MISR measurements agree on AOD increase by 0.001,

0.0007, and 0.0007/year respectively with correlation coefficients 0.10, 0.04, and 0.22

respectively. Despite the deviation between the three aforementioned sensors, they all agree on AOD temporal trend increase over Cairo. This could be attributed to the high pollution level at the mega city of Cairo due to high population, vehicle emission, and biomass burning.

Taman site (Fig. 4c): MISR AOD agrees with Taman AERONET on a positive trend indicating the efficiency of MISR V22 algorithm over green areas with less black carbon particles. Aqua measurements show temporal AOD decrease of 0.0079/year with a correlation coefficient of 0.81 and Terra show AOD decrease of 0.0043/year with a correlation coefficient of 0.35. Meanwhile, MISR shows AOD increase of 0.0014/year with a correlation coefficient of 0.19.

Long-range (2000 – 2015) tendency indicates that contradictory AOD trend of Terra and

Aqua is individually explicit for each site and does not necessarily apply everywhere.

AOD difference between Terra and Aqua could be used as another indicator of the long- range satellites performance. AOD difference (Terra AOD minus Aqua AOD) varies from -

0.01 to 0.19, -0.10 to 0.18, -0.02 to 0.13 over Solar Village, Taman, and Cairo respectively (Fig. 5). Over the Solar Village, Terra overestimates AODs during 2002-2004 and underestimates the AOD after 2005. Although Cairo and Taman show similar trend however over/underestimation amount is not unique for all sites. This is an indication that Aqua and

Terra retrievals disagreement takes place regardless of the region but site sampling has significant effect on the amount of contradiction.

Statistical comparison between MISR and MODIS/Terra AODs at corresponding

AERONET stations is performed by calculating Gfraction using of $\Delta\tau = \pm0.05 \pm 0.15\tau_{AERO}$

as a confidence interval. Over the region 1, MISR AODs retrievals are more accurate than

[revised manuscript text omitted]

in the matched data set but misses the peak at 0.5. Ångström exponent (AE), dependency of the AOD on wavelength, can also be used to determine particles' size where the smaller the particle the larger the exponent. AE analysis show that the first peak at 0.25 is indicative of industrial particles with high AE values and the second peak at 0.35 indicates dust aerosol.

The MISR climatology agrees well with the AERONET all data climatology for all AODs.

MODIS on the other hand shows an extremely large frequency of AODs at 0.1 not represented by AERONET coupled with an underestimation of AODs greater than 0.3. This could be attributed to the size of the matching window and MODIS retrievals preferentially coming from the Arabian Gulf.

SAADA station is located close to some hiking trails at the Agoundis Valley in the Atlas

Mountains about 197 km from the city of Marrakesh.

MISR AODs matched to AERONET agree well with MISR full climatology retrievals over

SAADA station. Both retrievals slightly underestimate SAADA full climatology and over estimates SAADA matched data retrievals at AODs equal to 0.1 while show good agreement for AODs greater than 0.1.  MODIS matched to AERONET retrievals overestimate the frequency of AODs greater than 0.3. While MODIS AODs matched to AERONET captures climatology at AODs between 0.2 to 0.25, AODs frequency retrievals are under-sampled at

AODs between 0.1 to 0.15 with about 13 % less events than SAADA all data retrievals at

AODs equal to 0.1.

Figure (9a, b) indicates right skewed distribution of SAADA AODs towards small AOD

values with 11.5 % and 30.1 % of AODs > 0.4 as measured by MISR and MODIS

respectively. Taking into consideration MODIS overestimation we conclude that SAADA

site is characterized by small AODs values and this could be related to the land topography where the station is located.

While MISR is capturing high AODs climatology over SAADA, both MISR and MODIS

are underestimating the frequency of lower AODs events. Nevertheless, MISR captures the climatology of AODs less than 0.1 missed by MODIS retrievals.

Taman AERONET station is located at the oasis city of Tamanrasset, which lies in Ahaggar

National Park at southern Algeria.

Figure (10 a, b) depicts that Taman AERONET AOD climatology is similar to those at

SAADA and has a high frequency of low AODs events. Both MISR AODs matched to

AERONET and MISR all data do not well capture the frequency of AODs less than 0.1 or larger than 1 while well describe the climatology for AODs in the range of 0.1 to 1. MODIS

AODs matched data to AERONET correctly describe climatology with slight overestimation of AODs frequencies between 0.05 – 0.15 while not capturing AODs frequencies greater than 1. MISR and MODIS show similar prominent peaks at 0.1, 0.25, and 0.35, not observed in Taman AERONET AOD climatology, with more peaks observed by MISR at 0.5, 0.6, and

0.8. Average AODs in SAADA and Taman is ~ 50 percent less than observed at Solar

Village, Mezaria, and Bahrain sites.

Except for AODs greater than 1 where ground observations could be more robust, both MISR

and MODIS retrievals can provide very good climatology matching over Taman site.

Taking into consideration lower number of MISR matching AERONET observations compared to MODIS ~ 33 and 43 percent over SAADA and Taman respectively, MISR is outperforming over these two sites, which can be attributed to its multiangle viewing capabilities over complex terrains including mountainous areas (Atlas Mountains).

Cairo is a mega city well known for its high pollution due to traffic and agriculture activities.

MISR and MODIS matched data correctly capture AOD climatology over Cairo compared to AERONET as shown in Figure (11a, b). MISR retrievals collocated with AERONET

capture prominent peaks of AERONET AOD at 0.15 – 0.25 and 0.5 with small underestimation observed at 0.3. MISR 'all data' AOD climatology over Cairo station agrees better with AERONET AOD climatology vs. collocated dataset with some oversampling at

0.15. Frequency of high AODs retrievals at 0.7 and 0.8 have not been captured by MISR

matched or all data retrievals. MODIS matched to AERONET AODs are also able to well present Cairo climatology data with a high overestimation of AODs frequency between 0.05

- 0.2 and an underestimation of AODs larger than 0.4.

[revised manuscript text omitted]

**Tables' caption**

Table 1. Geographic location of the AERONET sites used in this study

Table 2. Statistics for the calculation of MODIS/Terra, MODIS/Aqua, and MISR with that of AERONET measurements over seven sites in the Middle East and North Africa, including R: correlation coefficient, Gfraction: good fraction; N: number of observations

Table 3. Statistics for biomass and mixed sites, parameters as in Table 3. Caption.

Table 4. Percentage of AODs retrievals greater than 0.4 recorded by AERONET all data,

MISR all data and MODIS matched data over seven AERONET sites in Middle East and

North Africa.

**Figures caption**

Figure 1. Location of the AERONET stations over North Africa and the Middle East. The numbers on the map indicate the site location as 1: Saada, 2: Tamanrasset_INM, 3: Cairo,

4: Sede Boker, 5: Solar Village, 6: Mezaira, 7: Bahrain.

Figure 2. Scatter plot of MISR AOD versus AERONET AOD based on seasons and aerosols categorization.

Figure 3. Scatter plot of MODIS AOD versus AERONET AOD based on seasons and aerosols categorization.

Figure 4. Time series of monthly mean AOD derived from MODIS/Aqua, MODIS/Terra,

MISR and AERONET over a) dust b) biomass and c) mixed dominated aerosol regions.

Figure 5. Long range AOD difference for MODIS/Terra and MODIS/Aqua over the dust, biomass and mixed sites.

Figure 6. Histogram of the MISR, MODIS and Solar Village AERONET measurements a)

MISR b) MODIS data retrievals.

Figure 7. Histogram of the MISR, MODIS and Mezaria AERONET measurements a)

MISR b) MODIS data retrievals.

Figure 8. Histogram of the MISR, MODIS and Bahrain AERONET measurements a) MISR

b) MODIS data retrievals.

Figure 9. Histogram of the MISR, MODIS and SAADA AERONET measurements a)

MISR b) MODIS data retrievals.

Figure 10. Histogram of the MISR, MODIS and Taman AERONET measurements a)

MISR b) MODIS data retrievals.

Figure 11. Histogram of the MISR, MODIS and SEDEE Boker AERONET measurements a) MISR b) MODIS data retrievals.

Figure 12. Histogram of the MISR, MODIS and Cairo AERONET measurements a) MISR

b) MODIS data retrievals.

| Location name | Lon./Lat. | Measurement period |
|---|---|---|
| Solar Village | 24.907º N/46.397º E | 2000-2015 |
| Mezaria | 23.105º N/53.755º E | 2004-2015 |
| Bahrain | 26.208º N/50.609º E | 2000-2006 |
| Saada | 31.626º N/8.156º W | 2003-2015 |
| Taman | 22.790º N/5.530º E | 2000-2015 |
| Cairo | 30.081º N/31.290º E | 2005 -2007 |
| Sede Boker | 30.855º N/34.782 º E | 2000-2015 |

Table 1.

Table 2.

| AERONET Site | Sensor | Season | Mean Value | | N | R | Gfraction (%) |
|---|---|---|---|---|---|---|---|
| | | | AERONET | Satellite | | | |
| | MISR | DJF | 0.31±0.22 | 0.38±0.20 | 338 | 0.94 | 60.05 |
| | | MAM | 0.39±0.27 | 0.45±0.23 | 89 | 0.94 | 65.16 |
| | | JJA | 0.39±0.18 | 0.45±0.17 | 141 | 0.90 | 70.21 |
| | | SON | 0.27±0.16 | 0.35±0.14 | 3 | 0.99 | 33.33 |
| Solar Village | | DJF | 0.27±0.19 | 0.33±0.17 | 1500 | 0.48 | 51.80 |

| AERONET Site | | Season | Mean Value | | N | R | Gfraction (%) |
|---|---|---|---|---|---|---|---|
| | | | | | | | |
| | **MODIS Terra** | **MAM** | 0.36±0.24 | 0.26±0.17 | 389 | 0.68 | 90.23 |
| | | **JJA** | 0.34±0.17 | 0.42±0.19 | 429 | 0.41 | 54.31 |
| | | **SON** | 0.22±0.10 | 0.36±0.12 | 471 | 0.51 | 28.87 |
| **Mezaria** | **MISR** | **DJF** | 0.33±0.15 | 0.40±0.17 | 60 | 0.89 | 75.00 |
| | | **MAM** | 0.32±0.19 | 0.41±0.22 | 13 | 0.90 | 69.23 |
| | | **JJA** | 0.42±0.13 | 0.47±0.17 | 21 | 0.85 | 80.95 |
| | | **SON** | 0.29±0.07 | 0.36±0.07 | 22 | 0.87 | 77.27 |
| | **MODIS Terra** | **DJF** | 0.32±0.15 | 0.35±0.19 | 198 | 0.86 | 74.74 |
| | | **MAM** | 0.44±0.33 | 0.45±0.27 | 115 | 0.92 | 78.07 |
| | | **JJA** | 0.39±0.14 | 0.43±0.20 | 89 | 0.81 | 71.91 |
| | | **SON** | 0.28±0.13 | 0.30±0.16 | 97 | 0.87 | 77.31 |
| **Bahrain** | **MISR** | **DJF** | 0.37±0.11 | 0.31±0.10 | 17 | 0.73 | 100 |
| | | **MAM** | 0.31±0.11 | 0.28±0.14 | 3 | 0.89 | 100 |
| | | **JJA** | 0.40±0.09 | 0.36±0.09 | 8 | 0.69 | 100 |
| | | **SON** | 0.40±0.09 | 0.30±0.05 | 4 | 0.98 | 100 |
| | **MODIS Terra** | DJF | 0.42±0.29 | 0.20±0.19 | 121 | 0.41 | 93.38 |
| | | MAM | 0.50±0.28 | 0.13±0.15 | 25 | 0.26 | 96.00 |
| | | JJA | 0.55±0.26 | 0.31±0.27 | 42 | 0.50 | 88.09 |
| | | SON | 0.35±0.14 | 0.21±0.12 | 29 | 0.32 | 93.10 |

                                    Table 3.

| AERONET Site | Sensor | Season | Mean Value | | N | R | Gfraction (%) |
|---|---|---|---|---|---|---|---|
| | | | AERONET | Satellite | | | |
| | **MISR** | **DJF** | 0.24±0.16 | 0.22±0.15 | 149 | 0.93 | 97.29 |
| | | **MAM** | 0.21±0.13 | 0.19±0.11 | 53 | 0.89 | 96.15 |
| | | **JJA** | 0.29±0.14 | 0.27±0.15 | 80 | 0.93 | 97.46 |
| | | **SON** | 0.19±0.15 | 0.19±0.12 | 60 | 0.94 | 98.30 |

| | | | | | | | |
|---|---|---|---|---|---|---|---|
| **SAADA** | | **DJF** | 0.23±0.16 | 0.32±0.21 | 550 | 0.57 | 57.81 |
| | | **MAM** | 0.24±0.18 | 0.39±0.23 | 90 | 0.43 | 44.44 |
| | **MODIS** | **JJA** | 0.30±0.17 | 0.45±0.18 | 201 | 0.40 | 45.27 |
| | **Terra** | **SON** | 0.19±0.13 | 0.22±0.14 | 162 | 0.71 | 72.39 |
| **Taman** | | **DJF** | 0.19±0.23 | 0.24±0.19 | 135 | 0.92 | 70.89 |
| | | **MAM** | 0.29±0.22 | 0.35±0.24 | 24 | 0.97 | 82.60 |
| | **MISR** | **JJA** | 0.35±0.30 | 0.39±0.19 | 36 | 0.85 | 71.42 |
| | | **SON** | 0.19±0.15 | 0.19±0.12 | 60 | 0.94 | 98.30 |
| | | **DJF** | 0.19±0.22 | 0.18±0.16 | 319 | 0.67 | 81.81 |
| | **MODIS** | **MAM** | 0.24±0.19 | 0.22±0.17 | 67 | 0.55 | 83.58 |
| | **Terra** | **JJA** | 0.37±0.32 | 0.29±0.20 | 69 | 0.69 | 84.05 |
| | | **SON** | 0.14±0.14 | 0.13±0.10 | 117 | 0.54 | 84.61 |
| **Cairo** | | **DJF** | 0.33±0.20 | 0.28±0.11 | 13 | 0.94 | 100 |
| | | **MAM** | 0.22±0.06 | 0.24±0.08 | 5 | 0.99 | 100 |
| | **MISR** | **JJA** | 0.43±0.23 | 0.34±0.11 | 5 | 0.99 | 100 |
| | | **SON** | 0.38±0.21 | 0.29±0.12 | 4 | 0.97 | 100 |
| | | **DJF** | 0.33±0.16 | 0.20±0.11 | 158 | 0.30 | 95.56 |
| | **MODIS** | **MAM** | 0.32±0.16 | 0.12±0.08 | 39 | 0.25 | 100 |
| | **Terra** | **JJA** | 0.35±0.14 | 0.28±0.07 | 58 | 0.17 | 94.82 |
| | | **SON** | 0.38±0.19 | 0.20±0.09 | 29 | 0.07 | 93.82 |
| **SEDEE_BOKER** | | **DJF** | 0.14±0.06 | 0.21±0.07 | 23 | 0.87 | 40.90 |
| | | **MAM** | 0.14±0.05 | 0.24±0.09 | 13 | 0.68 | 33.33 |
| | **MISR** | **JJA** | 0.16±0.05 | 0.24±0.06 | 163 | 0.85 | 33.33 |
| | | **SON** | 0.15±0.07 | 0.23±0.06 | 72 | 0.89 | 33.80 |
| | | **DJF** | 0.16±0.12 | 0.23±0.14 | 1312 | 0.36 | 53.50 |
| | **MODIS** | **MAM** | 0.21±0.18 | 0.24±0.19 | 338 | 0.34 | 65.68 |
| | **Terra** | **JJA** | 0.16±0.09 | 0.33±0.13 | 392 | 0.27 | 17.34 |
| | | **SON** | 0.16±0.09 | 0.23±0.12 | 477 | 0.46 | 58.49 |

Table 4.

¶
¶
¶
¶
¶
¶
¶
¶
¶
¶
¶

| | AERONET | | MISR | | MODIS | |
|---|---|---|---|---|---|---|
| | | AOD | | AOD | | AOD |
| | N | % > 0.4 | N | % > 0.4 | N | % > 0.4 |
| Solar Village | 3978 | 28.7 | 684 | 32.8 | 2789 | 30.1 |
| Mezaria | 1650 | 30.2 | 547 | 45.7 | 498 | 40.7 |
| Bahrain | 1117 | 33.3 | 676 | 35.7 | 217 | 18.4 |
| SAADA | 3184 | 10.8 | 667 | 11.5 | 1004 | 34.6 |
| Taman | 1863 | 17.9 | 845 | 22.6 | 572 | 9.4 |
| Cairo | 269 | 53.5 | 620 | 17.7 | 284 | 4.2 |
| SEDEE | 5722 | 4.8 | 675 | 9 | 2519 | 12.8 |

[Figure]

Figure 1.

[Figure]

**Region 1**          **Region 2**

[Figure]

¶
¶
¶

[Figure]

                              Figure 2.

          **Region 1**                    **Region 2**

[Figure]

Figure 3.

[Figure]

Figure 4.

[Figure]

Figure 5.

[Figure]

[Figure]

Figure 6.

[Figure]

[Figure]

Figure 7.

[Figure]

[Figure]

Figure 8.

¶
¶

[Figure]

[Figure]

Figure 9.

[Figure]

[Figure]

Figure 10.

[Figure]

[Figure]

Figure 11.

[Figure]

[Figure]

Figure 12.

---

## Author Response (AR1)

Dear Dr. Marc Salzmann,

Thank you very much for your feedback regarding our article entitled ""Comparative Analysis of MODIS, MISR, and AERONET Climatology over the Middle East and North Africa". We greatly appreciate your comments and allowing extension for revising the manuscript.

I would also like to thank you for your encouraging us to include the latest MISR V23 data version in the results. As mentioned in my most recent email, I was able to obtain MISR V23 data (within 17.6 km) over 7 AERONET stations in the middle East and North Africa through Dr. Mike Gary at NASA Jet Propulsion Laboratory (JPL).

I have now repeated the analysis using MISR V23 version.

**Topical Editor Decision: Reconsider after major revisions (further review by editor and referees)** (01 Oct 2018) by Marc Salzmann
Comments to the Author:
Thank you very much for your replies to the reviewers' comments and to the comment by A.M. Sawyer. Based on the reviewers' suggestions and your response, I invite you to submit a revised version of your manuscript. This revised manuscript will most likely undergo further review by one of the two original reviewers. Please take into account the reviewers' suggestions when revising the manuscript.

**Topical Editor's comment**:
In the revised manuscript, the data versions should definitely be stated, preferably already in the first sentence of the abstract. I apologize for not noting this omission during my initial reading of the manuscript.

**Author's response**:

The data version has been included in the abstract.

**Topical Editor's comment**

In agreement with the comment by A. M. Sawyer, I also strongly encourage you to include the latest data versions. If needed, I would be glad to extend the deadline for potential revisions.

**Author's response**:

We have now included the most recent data version in the results. We truly appreciate your acceptance to our request of deadline extension.

**Topical Editor's comment**

On the other hand, when including the new versions, I would argue that it may be worthwhile to also retain the analysis of the older versions for comparison. After all, the older versions were released and used in other studies, and I would find it interesting to discuss the differences between the versions in the manuscript, independent of whether they are large or small, preferably in a separate section.

**Author's response**:

The main purpose of this article to compare data from two satellite instrument namely (MISR and MODIS on board of Terra satellite) with ground observation measured by AERONET stations in 7 locations in the Middle East and North Africa.

Although comparing different data versions of the same instrument could be a good idea but the authors respectfully feel that it will be out of the scope of the article. It will also make the article lengthy and confusing, as it will distract the reader from the main objective of the study.

That said, I have prepared a supplementary file that contains a comparison between different data version over the 7 AERONET stations. I have also included a detailed track changes version of the article so reviewer can see the changes we made using the new data.

If the reviewers and the topical editor still think that it is necessary to include results for different data version, we can still incorporate that in the paper, however the author do not recommend that.

**Topical Editor's comment**

If possible, the emphasis should, however, be on the latest versions since they are more relevant with respect to future studies. In case repeating the analysis with the new data versions is not feasible, please state in the manuscript that new versions are available.

**Author's response**:

Most recent data versions have been used in the manuscript.

**Comments from the author regarding using MISR V23 versus MISR V22 data**

As expected, we have found that the changes in MISR V23 new product has no significant impact on our results. The new product, however; performed better in describing the climatology at low AOD values and this could be attributed to the V23 added AOD grid points below 0.025, which eliminates gap at low AODs, observed relative to AERONET. This effect is obvious is AERONET stations like SADAA, Taman, Cairo, and Sedee Boker.

On the other hand, we have not seen a major difference between MISR V23 and V22 over the dust dominated AERONET stations like Solar Village and Mezaria. However, over Mezaria station V23 MISR matched AERONET data was found to better describe the climatology compared to V22. Over Bahrain, V23 MISR matched AERONET data did not capture few peaks at AOD between 0.25 – 0.4 but it successfully described the climatology between 0.45 – 0.60. The new version also over estimated AOD values larger than 0.6 over Bahrain.

For your consideration, we have included a copy of the revised article with track changes. Please find below our response to your comments. We also included a supplementary file to show the analysis using V22 and V23.

Please let me know if I can provide more information regarding the data used in the article. Thank you again for considering our article at Annales Geophysicae.

Ashraf Farahat

**Revision details of Manuscript "Comparative Analysis of MODIS, MISR, and AERONET Climatology over the Middle East and North Africa"**

Abstract:

P1 L2-3 added to indicate data level and version used in the manuscript.

2. Materials and Methods

2.1 MISR

P4 L 103 ver.0022 changed to ver.0023 to indicate using the latest MISR data.

4. Results and discussion

4.1 Validating MISR and MODIS AOD retrievals against AERONET observations over the Middle East and North Africa

P9 L292 new data have be placed based on MISR V23 data. 15 in DJF, 39 in MAM, 61 in JJA, and 23 in SON.

P10 L313 added (not shown in the figure).

P10 L318 "negative" has been replaced by "positive"

P11 L309 **68** has been replace by **64** based on MISR V23 data.

P11 L311 **72** has been replaced by **84** based on MISR V23 data.

4.3 Evaluating the MISR and MODIS climatology over Middle East and North Africa

P12 L322  0.55 has been replace by 0.50

P12 L323 added "that can be also observed in AERONET", added "that could not be"

P13 L336 added "Similar to the"

P13 L337 116 has been replaced by 213

P13 L338 1517 replaced by 2245

P 13 L341 0.3 replaced by 0.15.

P13 L342 added undersampled.

P13 L343 added more, added "matches the climatology", added "less than 2 percent"

P13 L347 added "to MISR data between 0.65 to 0.70 and at 0.35"

P13 L350 added "matched AERONET data" MISR also capture.

P14 L379 data have been updated to 0.20, 0.30, 0.45, 0.55, 0.7, 0.8, and 0.95

P14 L380 data have been updated to 0.55 and 0.70. AOD less than 0.15

P14 L382 data have been updated to 0.20, 0.30, and 0.35.

P14 L390-392 minor editing.

P14 L398 11.5 % changed to 10.3 % based on MISR V23 data.

P15 L 428 added "and"

P15 line 435 33 and 43 have been replaced by 21 and 49.

P15 line 441 added "estimate" and added "while underestimate"

P16 line 452 added AERONET AOD great than 0.35

P16 line 454 added "greater than"

P16 line 470 0.25 changed to 0.2

P 17 line 491 added "overestimate"

P17 lines 492 and 493 added "underestimate AOD > 0.4 over Cairo. MISR retrievals also match AOD > 0.4 for Mezaria and Sedee Boker",

P17 line 500 15.5 has been changed to 17.7 based on MISR V23 data

P17 line 503 15 has been changed to 13 based on MISR V23 data

P17 line 506 added "underestimating"

Conclusion

P18 and P19 conclusion has been edited based on the new revised data.

P30 and P31 Tables 2 and 3 have been edited using the new revised data

P33 Tables 4 has been edited using the new revised data

P35 Figure 2 has been updated using the new revised data

P36 Figure 3 has been updated using the new revised data

P37 Figure 4 has been updated using the new revised data.

P39- Figures 6, 7, 8, 9, 10, 11, and 12 have been updated using the new revised data.

**Re: Comparison between the analysis performed by MISR V22 and MISR V23**

Dear Dr. Marc Salzmann,

In this file, I tried to display a comparison between the analysis performed using MISR V22 (old version) and MISR V23 (latest version) data.

As the main purpose of this article to compare data from two-satellite instrument namely (MISR and MODIS on board of Terra satellite) with ground observation measured by AERONET stations in 7 locations in the Middle East and North Africa.

Although comparing different data versions of the same instrument could be a good idea but the authors respectfully feel that it will be out of the scope of the article. It will also make the article lengthy and confusing, as it will distract the reader from the main objective of the study.

That said, I have prepared this file to show the comparison between the analyses performed by the two MISR versions but I do not prefer to include it in the article.

If the reviewers and the topical editor still think that it is necessary to include results for different data version, we can still incorporate that in the paper, however the author do not recommend that.

Thank you again,

Ash

**Solar Village**

[Figure]

[Figure]

**Mezaria**

[Figure]

[Figure]

**Bahrain**

[Figure]

[Figure]

**SADAA**

[Figure]

a)

**MISR V22**

SAADA matched data $\tau$ = 0.237, $\sigma$ = 0.153, N = 338

MISR matched data $\tau$ = 0.219, $\sigma$ = 0.142, N = 338

SAADA all data $\tau$ = 0.200, $\sigma$ = 0.164, N = 3184

MISR all data $\tau$ = 0.205, $\sigma$ = 0.167, N = 667

Frequency of occurence %

AOD, 558 nm

[Figure]

a)

**MISR V23**

SAADA matched data $\tau$ = 0.192, $\sigma$ = 0.141, N = 214

MISR matched data $\tau$ = 0.195, $\sigma$ = 0.145, N = 214

SAADA all data $\tau$ = 0.195, $\sigma$ = 0.169, N = 2974

MISR all data $\tau$ = 0.207, $\sigma$ = 0.177, N = 447

Frequency of occurence %

AOD, 558 nm

**Taman**

[Figure]

[Figure]

**SEDEE Boker**

[Figure]

[Figure]

**Cairo**

[Figure]

a) **MISR V22**

Cairo matched data τ = 0.335, σ = 0.280, N = 22

MISR matched data τ = 0.280, σ = 0.115, N = 22

Cairo all data τ = 0.441, σ = 0.185, N = 269

MISR all data τ = 0.282, σ = 0.140, N = 620

a) **MISR V23**

Cairo matched data τ = 0.356, σ = 0.124, N = 138

MISR matched data τ = 0.284, σ = 0.124, N = 138

Cairo all data τ = 0.392, σ = 0.192, N = 2222

MISR all data τ = 0.284, σ = 0.123, N = 359

Dear Referee,

Thank you very much for your feedback about our article entitled "Comparative Analysis of MODIS, MISR, and AERONET Climatology over the Middle East and North Africa". We greatly appreciate the comments. We have addressed all your comments and the manuscript was revised accordingly.

In response to your comment regarding the comments of Dr. Andrew Sayer, NASA Goddard Space Flight Center, and as per the recommendation of Dr. Marc Salzmann the Topical editor regarding the MISR data version used in this study. We have revised the results based on the newest data available by MISR (V23).

As expected, we have found that the changes in MISR V23 new product has no significant impact on our results. The new product, however; performed better in describing the climatology at low AOD values and this could be attributed to the V23 added AOD grid points below 0.025, which eliminates gap at low AODs, observed relative to AERONET. This effect is obvious is AERONET stations like SADAA, Taman, Cairo, and Sedee Boker.

On the other hand, we have not seen a major difference between MISR V23 and V22 over the dust dominated AERONET stations like Solar Village and Mezaria. However, over Mezaria station V23 MISR matched AERONET data was found to better describe the climatology compared to V22. Over Bahrain, V23 MISR matched AERONET data did not capture few peaks at AOD between 0.25 – 0.4 but it successfully described the climatology between 0.45 – 0.60. The new version also over estimated AOD values larger than 0.6 over Bahrain.

For your consideration, we have included a copy of the revised article with track changes. Please find below our response to your comments. We also included a supplementary file to show the analysis using V22 and V23.

Please note that the first version of this response was received and published on Ann. Geophys. Discuss. on 19 September 2018.

Ashraf Farahat

Submitted on 24 Aug 2018
Anonymous Referee #1

Notes for the submission of interactive comments

**Anonymous: Yes** No

**Formal manuscript rating and recommendation to the editor** (non-public)

| | |
|---|---|
| Does the paper contain new data or new ideas or both of them? | Yes **No** |
| Are these up to international standards? | **Yes** No |
| Is the presentation clear? | **Yes** No |
| Does the author reach substantial conclusions? | Yes **No** |
| Is the length of the paper adequate? | **Yes** No |
| Is the language fluent and precise? | Yes **No** |
| Are the title and the abstract pertinent and understandable? | **Yes** No |
| Is the size of each figure adequate to the quantity of data it contains? | **Yes** No |

| Does the author give proper credit to related work and does he/she indicate clearly his/her own contribution? | **Yes** No | | |
|---|---|---|---|
| Would you cite this paper as a scientific contribution? | Very important  Fairly important | **May have potential after additional work and resubmission** | No potential value |

For final publication, the manuscript should be

**accepted as is**.

accepted subject to **technical corrections**.

accepted subject to **minor revisions**.

**reconsidered after major revisions:**

    I am willing to review the revised paper.

    **I am not willing to review the revised paper.**

**rejected**.

**Comments from Referees**

The histograms in figures 6-12 are the most useful depiction here of the collocated datasets, displaying the distributions of AOD values over the various AERONET sites as retrieved by AERONET and the satellite products. It is interesting that the MISR AOD retrieval does not appear to capture the very low AODs observed by AERONET. However the trend analysis provides a rather weak discussion and conclusion, only hinting at significant values for the Solar Village site with AERONET and MISR, as far as I can see from the figure.

**Author's response**

It is important to mention that the goal of this study is to assess the consistency in aerosol trends between spaceborne sensors and AERONET data. The study tried to investigate which satellite data can better describe ground-based measurements over certain geographic locations in the Middle East and North Africa. Our analysis mainly focused on how data availability, topography and water areas could affect satellite's measurements from one region to another. Aerosols categorization and sources are not the major focus of this study.

The following paragraphs have been added to section 4.2

p.10, lines 253-255 new paragraph added

Trends of aerosol loading from 2000 to 2005 are analysed by plotting fitting lines of monthly mean AOD retrievals by MISR and MODIS/Terra and Aqua. The AOD retrieved by different instrument shows different trends.

p.10, lines 257-260 new paragraph added

Terra depicts a negative correlation coefficient with time while Aqua shows a positive one. Terra AOD decreases 0.0071/year, while Aqua increases 0.0015/year. Aqua have lower correlation coefficient for AOD compared to Terra, which indicates Aqua performed more stable during the study period.

p.10, lines 265-267 new paragraph added

In order to understand whether the discrepancy temporal trend of Terra and Aqua is a result of regional conditions or if it exists in all sites, we investigated Terra, Aqua, MISR, and AERONET over other sites.

p.10, lines 268-270 paragraph has been modified

Both MODIS/Aqua and MODIS/Terra AOD show a stable trend over time at Mezaria site (not shown in the figure) with a correlation coefficient of 0.11 and 0.04 respectively. Both Terra and Aqua AOD increase 0.008 and 0.001/year, respectively.

p.10, 11, lines 272-282, new paragraph added where Terra AOD decreases 0.0027/year, while Aqua increases 0.0066/year. Although Solar Village, Mezaria, and Bahrain are all located in or next to a desert region, the inconsistency between Terra and Aqua measurements is subject to the regional conditions. For example, the large water body surrounding Bahrain could mean that the great majority of the MODIS retrievals are from Dark Target algorithm.

p.11, lines 291-295, new paragraph added indicating the efficiency of MISR V22 algorithm over green areas with less black carbon particles. Aqua measurements show temporal AOD decrease of 0.0079/year with a correlation coefficient of 0.81 and Terra show AOD decrease of 0.0043/year with a correlation coefficient of 0.35. Meanwhile, MISR shows AOD increase of 0.0014/year with a correlation coefficient of 0.19.

**Comments from Referees**

I noticed the short comment by Andrew Sayer (I usually try to avoid reading other reviews in discussion journals, but as a comment on data versions this seemed to be a particularly relevant point), and I agree that it is vital that the most up-to-date data versions are used for all three of the datasets. If the current versions are not used then the analysis in this paper is of only minimal historical interest. Therefore please make sure that you are using the new Version 3 AERONET products, for example. I do not know how much difference to the results re-performing the analysis will cause, but presumably there will be differences in almost all of the figures and tables.

**Author's response**

We would like to confirm that we have used Level 2.0 Version 3 AERONET data available at https://aeronet.gsfc.nasa.gov. This has been highlighted in the paper at p.6, lines 48-49.

For MODIS data, we have used Collection 6.1. Both dark target and deep blue algorithms have been used. Dark target retrievals were used over water regions while deep blue data were used over land. Data are available at https://giovanni.gsfc.nasa.gov/giovanni.

For MISR data, we have choose to use V22 rather than V23, released on February 12, 2018, in our analysis because of few know issues know with this product that are still under formal validation. Some of these known issues are directly related to data reliability over bright surfaces compared to dark water, which is significant for our study.

We have responded to Andrew Sayer through public discussion to explain that for the results reliability we should not use V23 MISR data for this study. Only after these known issues are resolved, it will be more feasible to relay on the new data product.

Below please find our detailed response to Andrew Sayer

Dear Andrew,

Thank you very much for the short comment regarding the data version used in the article.

**MISR**

Indeed, we are aware of version 23 (V23) MISR data released on February 12, 2018, however few known issues with the new product are still under formal validation. Some of these known issues are related to data reliability over bright surfaces compared to dark water, which is significant for our study.

Moreover, we have found that changes in the new product has no significant impact on the results presented in our article as explained below in major and minor differences between V23 and V22 MISR product.

To ensure data reliability based on known issues and insignificant impact of the new product on our results, we preferred to use the most recent V22 in our analysis.

**Major differences between V23 and V22 MISR products**

1- Initial assessments of the results from the 4.4 km resolution V23 retrieval algorithm show that V22 AOD retrievals perform similar to V23 relative to AERONET. V23, however perform significantly better than V22 only relative to high spatial density AERONET Distributed Regional Aerosol Gridded Observation Network (DRAGON) deployments which is out of the scope of our study.
2- V22 has similar performance as V23 in reporting non-spherical aerosols in places where they are climatologically expected, particularly when the AOD is large. Both versions effectively discriminates small, medium, and large particles in exactly similar pattern.
3- Although V23 added AOD grid points below 0.025, which eliminates gap at low AODs, observed relative to AERONET, this update should not affect the results in our article, as we are not dealing with such low AOD values.
4- V23 changes in the snow-ice mask source by applying a more conservative cloud screening logic. This should have no effect on the results presented in our paper as we have performed our comparative analysis mostly over an arid/semi-arid region.
5- V23 change in near-surface wind speed source has no significant effect on our results as only the total wind speed is used in the dark water aerosol retrievals; this change does not affect the Aerosol Product.
6- V23 added a correction factor to take into consideration the effect of chlorophyll ("underlight") on MISR red and NIR bands over Dark Water. This reduces AODs retrieved over dark water; however, its significantly affect low AODs values only.

**Minor differences between V23 and V22 MISR products**

1- Significant field name and content changes in V23 relative to V22, which makes the product significantly more accessible. This however has no effect on the results discussed in our article.
2- Switch from HDF4, stacked-block format to NetCDF-4 conventional format. This however has no effect on the results discussed in our article.

3- Provide per-retrieval geolocation and time information to make product easier to use. This also has no effect on the results presented.

If you still believe that the new data product could significantly change the results taking into consideration possible AOD range at the study region, please let me know and we can definitely check the results against the new version.

**AERONET**

For the AERONET data, we have used Level 2.0 Version 3 available at https://aeronet.gsfc.nasa.gov. We will highlight this in the article.

**MODIS**

For MODIS data, we have used Collection 6.1. Both dark target and deep blue algorithms have been used. Dark target retrievals were used over water regions while deep blue data were used over land. Data are available at https://giovanni.gsfc.nasa.gov/giovanni. We will highlight this in the article.

**Comments from Referees**

Please also clarify whether you are using the Dark Target (DT) and/or the Deep Blue (DB) AOD retrievals, since these use very different retrieval methods, and it is a vital distinction to make. Presumably the MODIS AODs over central desert sites such as Solar Village or Tamanrasset would be from the Deep Blue algorithm, while coastal sites such as Bahrain would have a greater prevalence of DT retrievals. It would perhaps make more sense to discriminate the MODIS AODs further, between retrievals using the DT and the DB algorithms. A possible question might be whether the DB or the DT algorithm performs better in the vicinity of Bahrain or other such sites on the desert margins?

**Author's response**

Both dark target and deep blue algorithms have been used. Dark target retrievals were used over water regions while deep blue data were used over land. Data are available at https://giovanni.gsfc.nasa.gov/giovanni.

For regions like Bahrain where large water body surrounds land, a combined Dark Target and Deep Blue AOD for land and Ocean has been used available https://giovanni.gsfc.nasa.gov/giovanni.

p.6, Lines 133- 137 was added.

**Specific Comments**

**Comments from Referees**

p.2, lines 36-37: why is this in italics?

**Author's response**

Italics format has been removed.

**Author's changes in manuscript**

p.2, lines 36-37: Italics format has been removed.

**Comments from Referees**

Section 2.2: if MODIS Deep Blue retrievals are used (and they should be), please also describe them here

**Author's response**

The author would like to confirm that both dark target and deep blue algorithms have been used. Dark target retrievals were used over water regions while deep blue data were used over land. Data are available at https://giovanni.gsfc.nasa.gov/giovanni.

The Deep Blue retrievals have been described on section 2.2 P5 L 117 - 124

The Deep Blue is a NASA developed algorithm to calculate AOD over land using MODIS data. Bu measuring contrast between aerosols and surface features, Deep Blue retrieves AOD. Over bright land, Deep Blue uses (0.412, 0.470/0.490 μm) and dark land (0.470/0.490, 0.650 μm) for AOD retrievals. Over water, the Deep Blue algorithm is not used.

The MODIS dark-target algorithm is designed aerosol retrieval from MODIS observations, over ocean (dark in visible and longer wavelengths) and dark land surfaces (low values of surface reflectance) (e.g., dark soil and vegetated regions) in parts of the visible (VIS, 0.47 and 0.65 μm) and shortwave infrared (SWIR, 2.1 μm) spectrum (Kaufman et al., 1997).

**Author's changes in manuscript**

New paragraph has been added to section 2.2 to describe Deep Blue algorithm P5 L 117 - 124

**Comments from Referees**

Throughout the manuscript there are language issues which should be corrected

**Author's changes in manuscript**

**Comments from Referees**

p.14, line 330: do you know what these peaks indicate? On brief speculation I might imagine that the first peak is indicative of industrial aerosol and the second peak might be indicative of dust. Ångström coefficient values may give some evidence as to what these might be.

**Author's response**

**P14, lines 332-335 have been added**

Ångström exponent (AE), dependency of the AOD on wavelength, can also be used to determine particles' size where the smaller the particle the larger the exponent. AE analysis show that the first peak at 0.25 is indicative of industrial particles with high AE values and the second peak at 0.35 indicates dust aerosol. High anthropogenic loading could be attributed to rapidly growing aluminum industry in Bahrain (Farahat 2016).

**Comments from Referees**

p.14, lines 337-338: if the MODIS retrievals are preferentially coming from the Gulf, does that mean that the great majority of the retrievals over Bahrain are from DT?

**Author's response**

The MODIS matched AERONET data are averaged from measurements that are within a radius of about 27.5 km from the AERONET station and within 30 min of the satellite flyover the station. For such a small country like Bahrain surrounded with a large water area, MODIS retrievals are preferentially coming from the water. Combined Dark Target and Deep Blue products are used for Bahrain the majority of the measurement are from DT.

**Comments from Referees**

p.14, line 253: 'topology'. I think you mean 'topography'?

**Author's response**

Thank you. 'topology' has been replaced with 'topography'

**Author's changes in manuscript**

p.15, line 254

**Additional revision details (after major revision)**

**Manuscript "Comparative Analysis of MODIS, MISR, and AERONET Climatology over the Middle East and North Africa"**

Abstract:

P1 L2-3 added to indicate data level and version used in the manuscript.

2. Materials and Methods

2.1 MISR

P4 L 103 ver.0022 changed to ver.0023 to indicate using the latest MISR data.

4. Results and discussion

4.1 Validating MISR and MODIS AOD retrievals against AERONET observations over the Middle East and North Africa

P9 L292 new data have be placed based on MISR V23 data. 15 in DJF, 39 in MAM, 61 in JJA, and 23 in SON.

P10 L313 added (not shown in the figure).

P10 L318 "negative" has been replaced by "positive"

P11 L309 **68** has been replace by **64** based on MISR V23 data.

P11 L311 **72** has been replaced by **84** based on MISR V23 data.

4.3 Evaluating the MISR and MODIS climatology over Middle East and North Africa

P12 L322  0.55 has been replace by 0.50

P12 L323 added "that can be also observed in AERONET", added "that could not be"

P13 L336 added "Similar to the"

P13 L337 116 has been replaced by 213

P13 L338 1517 replaced by 2245

P 13 L341 0.3 replaced by 0.15.

P13 L342 added undersampled.

P13 L343 added more, added "matches the climatology", added "less than 2 percent"

P13 L347 added "to MISR data between 0.65 to 0.70 and at 0.35"

P13 L350 added "matched AERONET data" MISR also capture.

P14 L379 data have been updated to 0.20, 0.30, 0.45, 0.55, 0.7, 0.8, and 0.95

P14 L380 data have been updated to 0.55 and 0.70. AOD less than 0.15

P14 L382 data have been updated to 0.20, 0.30, and 0.35.

P14 L390-392 minor editing.

P14 L398 11.5 % changed to 10.3 % based on MISR V23 data.

P15 L 428 added "and"

P15 line 435 33 and 43 have been replaced by 21 and 49.

P15 line 441 added "estimate" and added "while underestimate"

P16 line 452 added AERONET AOD great than 0.35

P16 line 454 added "greater than"

P16 line 470 0.25 changed to 0.2

P 17 line 491 added "overestimate"

P17 lines 492 and 493 added "underestimate AOD > 0.4 over Cairo. MISR retrievals also match AOD > 0.4 for Mezaria and Sedee Boker",

P17 line 500 15.5 has been changed to 17.7 based on MISR V23 data

P17 line 503 15 has been changed to 13 based on MISR V23 data

P17 line 506 added "underestimating"

Conclusion

P18 and P19 conclusion has been edited based on the new revised data.

P30 and P31 Tables 2 and 3 have been edited using the new revised data

P33 Tables 4 has been edited using the new revised data

P35 Figure 2 has been updated using the new revised data

P36 Figure 3 has been updated using the new revised data

P37 Figure 4 has been updated using the new revised data.

P39- Figures 6, 7, 8, 9, 10, 11, and 12 have been updated using the new revised data.

Dear Referee,

Thank you very much for your comments regarding our article entitled "Comparative Analysis of MODIS, MISR, and AERONET Climatology over the Middle East and North Africa". We greatly appreciate the feedback. We have addressed all your comments and we have accoringly revised the manuscript.

In addition to your comments, we have also revised the manuscript based on the comments of Dr. Andrew Sayer, NASA Goddard Space Flight Center, and the recommendation of Dr. Marc Salzmann the Topical editor regarding the MISR data version used in this study. We have revised the results based on the newest data available by MISR (V23).

As expected, we have found that the changes in MISR V23 new product has no significant impact on our results. The new product, however; performed better in describing the climatology at low AOD values and this could be attributed to the V23 added AOD grid points below 0.025, which eliminates gap at low AODs, observed relative to AERONET. This effect is obvious is AERONET stations like SADAA, Taman, Cairo, and Sedee Boker.

On the other hand, we have not seen a major difference between MISR V23 and V22 over the dust dominated AERONET stations like Solar Village and Mezaria. However, over Mezaria station V23 MISR matched AERONET data was found to better describe the climatology compared to V22. Over Bahrain, V23 MISR matched AERONET data did not capture few peaks at AOD between 0.25 – 0.4 but it successfully described the climatology between 0.45 – 0.60. The new version also over estimated AOD values larger than 0.6 over Bahrain.

For your consideration, we have included a copy of the revised article with track changes. Please find below our response to your comments. We also included a supplementary file to show the analysis using V22 and V23.

Please note that the first version of this response was received and published on Ann. Geophys. Discuss. on 19 September 2018.

**Anonymous: Yes** No

**Formal manuscript rating and recommendation to the editor** (non-public)

| | |
|---|---|
| Does the paper contain new data or new ideas or both of them? | **Yes** No |
| Are these up to international standards? | **Yes** No |
| Is the presentation clear? | **Yes** No |
| Does the author reach substantial conclusions? | **Yes** No |
| Is the length of the paper adequate? | **Yes** No |
| Is the language fluent and precise? | Yes **No** |
| Are the title and the abstract pertinent and understandable? | **Yes** No |
| Is the size of each figure adequate to the quantity of data it contains? | Yes **No** |
| Does the author give proper credit to related work and does he/she | **Yes** No |

| indicate clearly his/her own contribution? | | | | |
|---|---|---|---|---|
| Would you cite this paper as a scientific contribution? | Very important | Fairly important | **May have potential after additional work and resubmission** | No potential value |

For final publication, the manuscript should be
**accepted as is**.
accepted subject to **technical corrections**.
**accepted subject to minor revisions.**
reconsidered after **major revisions**:
    I am willing to review the revised paper.
    I am **not** willing to review the revised paper.
**rejected**.

**Comments from Referees**

The author of this manuscript has done quite interesting work, well analyzed "Comparative Analysis of MODIS, MISR 1 and AERONET Climatology over the Middle East and North Africa". In general the manuscript is interesting and well written. The results have been presented and discussed well and thoroughly. In my opinion, the topic discussed in this paper is suitable for publication. Overall I recommend acceptance of this paper for publication with minor revisions. Please see the specific comments below.

**Author's response**

We would like to thank the reviewer very much for his/her comments and for recommending the publication of our article with minor revision. We have addressed all the reviewer comments below.

**Comments from Referees**

Line 11: please insert comma after MISR

**Author's response**

Done

**Comments from Referees**

Line 15: please check the grammar, i.e. MODIS/terra AOD indicates instead of indicate

**Author's response**

Done

**Comments from Referees**

Line 33: please use like this "that has major effects on human activities in the Arabian"

**Author's response**

Done

**Comments from Referees**

Line 42-43: please make it clear to the reader

**Author's response**

p.2 Lines 42-43 have been modified

Aerosol optical depth, AOD, is a parameter to measure the extinction of a beam of light as it passes through a layer of atmosphere that contains aerosols.

**Comments from Referees**

Line 121: please rephrase the sentence.

**Author's response**

p. 5 Lines 126-129 (previous 121 – 124) have been rephrased.

The MODIS dark-target algorithm derives aerosol characteristics, including AOD, over ocean (dark in visible and longer wavelengths) and dark land surfaces (low values of surface reflectance) (e.g., dark soil and vegetated regions) in parts of the visible (VIS, 0.47 and 0.65 μm) and shortwave infrared (SWIR, 2.1 μm) spectrum (Kaufman et al., 1997).

**Comments from Referees**

Line 136-137: please rephrase the sentence

**Author's response**

p. 6 Lines 152-154 (previous 136 -138) have been rephrased

The sun photometers used by AERONET include sun collimators to measure spectral direct-beam solar radiation. The collimators are used to determine columnar spectral AOD and water vapour, provided at a temporal resolution of approximately 10–15 min (Sayer et al. 2014).

**Comments from Referees**

Line 142: please mention the name of satellite

**Author's response**

The names of the satellites are now mentioned p.6 L157-158 (previous L 142)

Seven AERONET sites were selected for MODIS/ Terra, MODIS/ Aqua, and MISR/Terra satellites validation in this study (Table 1.).

**Comments from Referees**

Line 147-149: please revise the sentence.

**Author's response**

p.7 Lines 170 – 172 have been revised (previous 147-149).

Multi-sensors data matching approach requires using only spatial and temporal matching data to reduce uncertainties associated with using different instruments and clouds shadow Liu and Mishchenko (2008) and Mishchenko et al., 2009.

**Comments from Referees**

Line 158: The authors have mentioned that they have used second approach in this study. Why did the authors not use the first approach?

**Author's response**

Both approaches have their limitations; however, we used (Mishchenko et al., 2010 approach) as it simultaneously matches location and time between the AERONET station and satellites. This certainly reduces the number of available matched data points; however, it eliminates data uncertainty compared to the other approach.

**Comments from Referees**

Line 176: The authors have used only two statistics parameters to validate the satellite data. It is suggested to use more parameters for the validation. It is also observed that authors have not mention the value of statistical parameters in the figures.

**Author's response**

We totally agree with the referee comments that more statistical parameters would strength the validation process. Indeed, we have tried to use fours statistical parameters namely relative error, correlation coefficient, root mean square deviation, and good fraction. That said, for our specific study we found that the same conclusion can be approached using only two parameters. In order to avoid lengthy tables and redundancy that may confuse readers, we decided to present two parameters only in the tables.

We have presented some of the statistical parameters in the figures, the rest are listed in Tables 1-4.

**Comments from Referees**

Line 196: please correct number of equations in the text.

**Author's response**

**p.8 Lines 224, and 225. (Previous Line 196).**

Thank you. Equation numbers are now corrected.

**Comments from Referees**

Table 2: Caption of table should be precise and general and table value should match according to the caption e.g RMSE is mentioned in the caption but not presented in the table, G-fraction and Gfraction should same in the text.

**Author's response**

Table 2 caption has been modified p.28 Lines 838-840

Table 2. Statistics for the calculation of MODIS/Terra, MODIS/Aqua, and MISR with that of AERONET measurements over seven sites in the Middle East and North Africa, including R: correlation coefficient, Gfraction: good fraction; N: number of observations

We have also used "Gfraction" all over the text.

**Comments from Referees**

Table 3: Like statistics for biomass and mixed, parameter as in table 2 (but you mentioned parameter as table 3)

**Author's response**

Thank you. Typo corrected. P.28 Lines 782

**Comments from Referees**

Second column of each table should be same if they belongs to same category. It will confuse the reader, like in table 2, you used 'sensor' but in table 3 you changed sensor to 'method' but they are the same indeed. It will confuse the reader

**Author's response**

Thank you. "Method" has been changed to "Sensor" in Table 3 Column 2

**Comments from Referees**

Table 4: Caption of table 4 is again confusing MISR coverage but in the body of table MODIS, MISR and AERONET are all showing their coverage

**Author's response**

Thank you. Table 4 caption has been modified to

Table 4. Percentage of AODs retrievals greater than 0.4 recorded by AERONET all data, MISR all data and MODIS matched data over seven AERONET sites in Middle East and North Africa.

**Comments from Referees**

FIGURE 1: Check the grammar of caption of figure1 e.g. "The numbers on the map indicate, not indicates" What is the source of this fig? Please combine figure 2 and 3 because they are the same actually just with different satellite data

**Author's response**

We have corrected the grammar of figure 1 caption.

We have produced the map in figure 1 in house using GIS software.

We would like to thank the reviewer for his/her suggestion of combining figure 2 and figure 3 but we respectfully prefer to keep them as separate figures. Combining the two figures will make them not clear.

**Additional revision details (after major revision)**

**Manuscript "Comparative Analysis of MODIS, MISR, and AERONET Climatology over the Middle East and North Africa"**

Abstract:

P1 L2-3 added to indicate data level and version used in the manuscript.

2. Materials and Methods

2.1 MISR

P4 L 103 ver.0022 changed to ver.0023 to indicate using the latest MISR data.

4. Results and discussion

4.1 Validating MISR and MODIS AOD retrievals against AERONET observations over the Middle East and North Africa

P9 L292 new data have be placed based on MISR V23 data. 15 in DJF, 39 in MAM, 61 in JJA, and 23 in SON.

P10 L313 added (not shown in the figure).

P10 L318 "negative" has been replaced by "positive"

P11 L309 **68** has been replace by **64** based on MISR V23 data.

P11 L311 **72** has been replaced by **84** based on MISR V23 data.

4.3 Evaluating the MISR and MODIS climatology over Middle East and North Africa

P12 L322  0.55 has been replace by 0.50

P12 L323 added "that can be also observed in AERONET", added "that could not be"

P13 L336 added "Similar to the"

P13 L337 116 has been replaced by 213

P13 L338 1517 replaced by 2245

P 13 L341 0.3 replaced by 0.15.

P13 L342 added undersampled.

P13 L343 added more, added "matches the climatology", added "less than 2 percent"

P13 L347 added "to MISR data between 0.65 to 0.70 and at 0.35"

P13 L350 added "matched AERONET data" MISR also capture.

P14 L379 data have been updated to 0.20, 0.30, 0.45, 0.55, 0.7, 0.8, and 0.95

P14 L380 data have been updated to 0.55 and 0.70. AOD less than 0.15

P14 L382 data have been updated to 0.20, 0.30, and 0.35.

P14 L390-392 minor editing.

P14 L398 11.5 % changed to 10.3 % based on MISR V23 data.

P15 L 428 added "and"

P15 line 435 33 and 43 have been replaced by 21 and 49.

P15 line 441 added "estimate" and added "while underestimate"

P16 line 452 added AERONET AOD great than 0.35

P16 line 454 added "greater than"

P16 line 470 0.25 changed to 0.2

P 17 line 491 added "overestimate"

P17 lines 492 and 493 added "underestimate AOD > 0.4 over Cairo. MISR retrievals also match AOD > 0.4 for Mezaria and Sedee Boker",

P17 line 500 15.5 has been changed to 17.7 based on MISR V23 data

P17 line 503 15 has been changed to 13 based on MISR V23 data

P17 line 506 added "underestimating"

Conclusion

P18 and P19 conclusion has been edited based on the new revised data.

P30 and P31 Tables 2 and 3 have been edited using the new revised data

P33 Tables 4 has been edited using the new revised data

P35 Figure 2 has been updated using the new revised data

P36 Figure 3 has been updated using the new revised data

P37 Figure 4 has been updated using the new revised data.

P39- Figures 6, 7, 8, 9, 10, 11, and 12 have been updated using the new revised data.

[revised manuscript text omitted]

Margin annotations (tracked changes):
- **Deleted:** 97.29
- **Deleted:** 96.15
- **Deleted:** 97.46
- **Deleted:** 98.30
- **Deleted:** 1
- **Deleted:** 4
- **Deleted:** 7
- **Deleted:** 9
- **Deleted:** 70.89
- **Deleted:** 2.60
- **Deleted:** 1.42
- **Deleted:** 8.30
- **Deleted:** 1
- **Deleted:** 8
- **Deleted:** 5
- **Deleted:** 1
- **Deleted:** 100
- **Deleted:** 100
- **Deleted:** 100
- **Deleted:** 6
- **Deleted:** 2
- **Deleted:** 2

| | | | | | | | |
|---|---|---|---|---|---|---|---|
| | MISR | DJF | 0.11±0.06 | 0.13±0.05 | 10 | 0.87 | 90.0 |
| | | MAM | 0.21±0.13 | 0.24±0.13 | 76 | 0.68 | 75.0 |
| | | JJA | 0.16±0.08 | 0.21±0.08 | 142 | 0.85 | 66.9 |
| SEDEE_BOKER | | SON | 0.162±0.07 | 0.20±0.06 | 54 | 0.89 | 79.6 |
| | MODIS Terra | DJF | 0.16±0.12 | 0.23±0.14 | 1312 | 0.36 | 53.5 |
| | | MAM | 0.21±0.18 | 0.24±0.19 | 338 | 0.34 | 65.6 |
| | | JJA | 0.16±0.09 | 0.33±0.13 | 392 | 0.27 | 17.3 |
| | | SON | 0.16±0.09 | 0.23±0.12 | 477 | 0.46 | 58.4 |

                          Table 4.

|  | AERONET | | MISR | | MODIS | |
|---|---|---|---|---|---|---|
|  |  | AOD |  | AOD |  | AOD |
|  | N | % > 0.4 | N | % > 0.4 | N | % > 0.4 |
| Solar Village | 3893 | 27.17 | 684 | 32.8 | 2789 | 30.1 |
| Mezaria | 2245 | 28.01 | 547 | 45.7 | 498 | 40.7 |
| Bahrain | 1116 | 31.36 | 676 | 35.8 | 217 | 18.4 |
| SAADA | 2974 | 10.32 | 667 | 11.5 | 1004 | 34.6 |
| Taman | 798 | 15.78 | 845 | 22.6 | 572 | 9.4 |
| Cairo | 2222 | 38.79 | 620 | 17.7 | 284 | 4.2 |
| SEDEE | 5722 | 4.28 | 675 | 9.0 | 2519 | 12.8 |

[Figure]

Figure 1.

¶
¶
¶
¶
¶
¶
¶
¶
¶
¶

Region 1                                    Region 2

[Figure]

                                Figure 2

[Figure]

Figure 3

¶
¶
¶
¶
¶
¶
¶
¶
¶
¶
¶
¶
¶
¶

[Figure]

 Figure 4.

Moved up [1]: Figure 2.¶

¶
¶
¶
¶
¶
¶

Moved (insertion) [2]

¶
¶
¶
¶
¶
¶
¶
¶
¶
¶
¶
¶
¶
¶
¶
¶
¶
¶

Moved up [2]: Figure 3.¶

¶
¶
*<object>*¶
¶
¶
¶
¶
*<object>*¶
¶
¶
¶
¶
*<object>*¶
¶
¶
¶
¶
¶
¶
¶

[Figure]

Figure 5.

[Figure]

[Figure]

[Figure]

[Figure]

Figure 6.

[Figure]

[Figure]

[Figure]

[Figure]

Figure 8.

¶
*<object>*

[Figure]

[Figure]

[Figure]

Figure 9.

¶
¶

[Figure]

[Figure]

[Figure]

[Figure]

Figure 10.

¶
¶

[Figure]

[Figure]

[Figure]

[Figure]

Figure 11.

[Figure]

[Figure]

[Figure]

Figure 12.

¶

---

## Author Response (AR2)

Dear Dr. Marc Salzmann, Topical editor,

Thank you very much for your feedback regarding our manuscript entitled "Comparative Analysis of MODIS, MISR, and AERONET Climatology over the Middle East and North Africa". We greatly appreciate your comments. We have addressed all your comments and revised the manuscript accordingly.

Please find below a point-by-point response to your comments.

For your consideration, we have included a copy of the revised article with track changes.

Kindly let me know if I can provide more details regarding the manuscript.

Thank you,

Ashraf Farahat

**Reviewers' comment**

l. 95 to measures -> measures

**Response**

Done

**Reviewers' comment**

l. 110: Terra's -> Terra

**Response**

Done (now l.111 p5)

**Reviewers' comment**

l. 117 provides -> provide

**Response**

Done (now l.118 p5)

**Reviewers' comment**

l. 120 designed aerosol -> designed for aerosol

**Response**

Done (now l. 126 p5)

**Reviewers' comment**

l. 125 for (C005) ->? Please clarify

**Response**

l.131 p5: (C005) modified to Collection 5 (C005), also l.129 p5 Level 2 (C006) modified to Level 2 – Collection 6 (C006)

**Reviewers' comment**

l. 126: land respectively -> land, respectively.

**Response**

Done (now l. 136 p6)

**Reviewers' comment**

l. 197: A good aspect -> An advantage

**Response**

Done (now l. 126 p9)

**Reviewers' comment**

l. 211: less error -> smaller deviations from the AERONET data

**Response**

Done (now l. 228 p9)

**Reviewers' comment**

l. 226f: either "than those" -> "than that" or else: "daily variability ... s" -> "daily variabilities ... are"

**Response**

Done "than those" -> "than that" (now l. 246 p10)

**Reviewers' comment**

l. 227: at -> in

**Response**

Done (now l. 246 p10)

**Reviewers' comment**

l. 264 and also in lines 267, 436, and 808: Long-range -> Long-term

**Response**

Done

**Reviewers' comment**

l. 275: omit "of"

**Response**

Done (now l. 299 p12)

**Reviewers' comment**

l. 289: illustrated -> shown

**Response**

Done (now l. 313 p12)

**Reviewers' comment**

l. 318: capture -> captures

**Response**

Done (now l. 344 p14)

**Reviewers' comment**

l. 379: present -> represent

**Response**

Done (now l.390 p 16)

**Reviewers' comment**

l. 425: Can -> Can be

**Response**

Done (now l.436 p 18)

**Reviewers' comment**

l. 425: atmosphere -> the atmosphere

**Response**

Done (now l.436 p 18)

**Reviewers' comment**

l. 479: while show -> while they show

**Response**

Done (now l.481 p 20)

**Reviewers' comment**

l. 480: do a good job -> perform well

**Response**

Done (now l.491 p20)

**Reviewers' comment**

l. 481: present -> represent

**Response**

Done (now l.508 p20)

**Reviewers' comment**

The following point regarding MODIS from the author's response should be clarified in the materials and methods section:

Both dark target and deep blue algorithms have been used. Dark target retrievals were used over water regions while deep blue data were used over land. For regions like Bahrain where large water body surrounds land, a combined Dark Target and Deep Blue AOD for land and Ocean has been used.

**Response**

**Lines 121-125 P5 have been added to the manuscript**

The Deep Blue (DB) is a NASA developed algorithm to calculate AOD over land using MODIS data. By measuring contrast between aerosols and surface features, DB retrieves AOD. Over bright land, DB uses (0.412, 0.470/0.479 µm) for AOD retrievals. Over water, the DB algorithm is not used, but the Dark Target (DT) algorithm is used instead.

**Lines 138-144 P6 have been also added to the manuscript**

Both DB and DT algorithms have been used in this study. DB data were used over land, while DT retrievals were used over water. For regions like Bahrain where large water body surrounds land, a combined DB and DT algorithm for land and ocean has been used. This is because the MODIS matched ground-based AERONET station in Bahrain (described in section 2.3 and Table 1) is located less than 2 km from the coastline. This makes MODIS combine retrievals for both land and water over this station. Data are available at https://giovanni.gsfc.nasa.gov/giovanni.

**Reviewers' comment**

As far as I can see, the last point raised by reviewer #1 was not answered. Table 1 says that AERONET data for Cairo is available for 2005 -2007, but this is inconsistent with Fig. 4b and the newly added text. Is the data displayed in Figure 3b available from the AERONET website? Please re-check!

**Response**

I agree with the reviewer's comment and I am very sorry, as I have overlooked that comment in my last response. There are two main sets of AERONET data available over Cairo; namely Cairo

EMA and Cairo EMA_2. Data listed under Cairo EMA are available from 2005 – 2007 only, while data listed under Cairo EMA_2 are available from 2010 – 2017.

In this study, we are comparing satellites' retrievals with AERONET station data using measurement that are within a radius of ~ 27.5 km from the AERONET station and about 30 min of each satellite flyover the AERONET location. For the Cairo station, we have these satellite matching data available for the Cairo EMA_2 retrievals; that is why we have used these data in Figure 4b.

We have now modified table 1 for Cairo AERONET station to show that the data available are from 2010-2017, which now matches figure 4b.

All AERONET data are available through https://aeronet.gsfc.nasa.gov

**More details**: to download data for the Cairo AERONET station

- Visit https://aeronet.gsfc.nasa.gov
- Under AEROSOL OPTICAL DEPTH (V3) click on Download Tool.
- Under Geographic Location pick "North_Africa" then click on "Get Country or State"
- Pick Egypt then click on Get AERONET Sites and pick Cairo_EMA_2.
- Click on Get Download Form.
- Data are available until 26-04-2010 to 14-03-2017

Note: AERONET satellite matching data are available by request through Jet Propulsion Laboratory (JPL)/NASA.

[revised manuscript text omitted]

                              Figure 2                                    Moved (insertion) [1]

**Region 1**          **Region 2**

[Figure]

Figure 3.

[Figure]

[Figure]

[Figure]

Figure 4.

[Figure]

Figure 5.

[Figure]

[Figure]

Figure 6.

[Figure]

[Figure]

Figure 7.

¶

[Figure]

[Figure]

Figure 8.

[Figure]

[Figure]

Figure 9.

[Figure]

[Figure]

Figure 10.

[Figure]

[Figure]

Figure 11.

[Figure]

[Figure]

Figure 12.